

# Contrasting composition of terrigenous organic matter in the dissolved, particulate and sedimentary organic carbon pools on the outer East Siberian Arctic Shelf

Joan A. Salvadó[1,2], Tommaso Tesi[1,2,3], Marcus Sundbom[1,2], Emma Karlsson[1,2],
Martin Kruså[1,2], Igor P. Semiletov[4,5,6], Elena Panova[4], Örjan Gustafsson[1,2]
[1]Department of Environmental Science and Analytical Chemistry (ACES), Stockholm
University, SE-10691 Stockholm, Sweden
[2]Bolin Centre for Climate Research, Stockholm University, SE-10691 Stockholm, Sweden
[3]CNR-National Research Council of Italy, ISMAR-Marine Sciences Institute in Bologna, Via
P. Gobetti 101, 40129 Bologna, Italy
[4] Tomsk Polytechnic University, Tomsk, Russia
[5]Pacific Oceanological Institute, Russian Academy of Sciences Far Eastern Branch,
Vladivostok 690041, Russia
[6]International Arctic Research Center, University Alaska Fairbanks, Fairbanks, AK 99775,
USA




**Abstract**

23        Fluvial discharge and coastal erosion of the permafrost-dominated East Siberian

Arctic delivers large quantities of terrigenous organic carbon (Terr-OC) to marine waters.
The composition and fate of the remobilized Terr-OC needs to be better constrained as it
impacts the potential for a climate-carbon feedback. In the present study, the bulk isotope
($\delta^{13}$C and $\Delta^{14}$C) and macromolecular (lignin-derived phenols) composition of the cross-shelf
exported organic carbon (OC) in different marine pools is evaluated. For this purpose, as part
of the SWERUS-C3 expedition (July-September 2014), sediment organic carbon (SOC) as
well as water column (from surface and near-bottom seawater) dissolved organic carbon
(DOC) and particulate organic carbon (POC) samples were collected along the outer shelves
of the Kara Sea, Laptev Sea and East Siberian Sea. The results show that the Lena River and
the DOC have a preferential role in the transport of Terr-OC to the outer shelf. DOC
concentrations (740-3600 µg/L) were one order of magnitude higher than POC (20-360
µg/L), with higher concentrations towards to the Lena River plume. Depleted $\delta^{13}$C, modern
$\Delta^{14}$C and lignin phenols concentrations were all well correlated with DOC levels indicating a
relatively young terrestrial contribution. In contrast, POC may have a preferential marine
origin, as its concentrations were not correlated with isotope and terrestrial biomarker
proxies. The $\delta^{13}$C signatures in the three carbon pools varied from -23.9±1.9‰ in the SOC, -
26.1±1.2‰ in the DOC and -27.1±1.9‰ in the POC. The $\Delta^{14}$C values ranged between -
395±83‰ (SOC), -226±92‰ (DOC) and -113±122‰ (POC). These stable and radiocarbon
isotopes were also different between the Laptev Sea and the East Siberian Sea. Both DOC
and POC showed a depleted and younger trend off the Lena River plume. The older and more
enriched $\delta^{13}$C signatures in the outer-shelf of the ESS suggest instead a greater influence of
the sea ice coverage and the Pacific inflow from the east. Lignin phenols exhibited higher
OC-normalized concentration in the SOC (0.10-2.34 mg/g OC) and DOC (0.08-2.40 mg/g
OC) than in the POC (0.03-1.14 mg/g OC). The good relationship between lignin and $\Delta^{14}$C
signatures in the DOC suggests that a significant fraction of the outer-shelf DOC comes from
"young" Terr-OC. By contrast, the slightly negative correlation between lignin phenols and
$\Delta^{14}$C signatures in POC, with higher lignin concentrations in older POC from near-bottom
waters, may reflect the off-shelf transport of OC from remobilized permafrost in the
nepheloid layer. Moreover, syringyl/vanillyl and cinnamyl/vannillyl phenols ratios presented
distinct clustering between DOC, POC and SOC, suggesting that those pools are carrying
different Terr-OC of partially different origin.  Finally, 3,5-dihydroxybenzoic acid to vanillyl



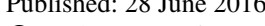


phenols ratios and p-coumaric acid to ferulic acid ratios, used as a diagenetic indicators,
enhanced in POC and SOC. This suggests that the remobilized old OC from thawing
permafrost, which is mainly transported within these pools, could experience less burial and
more mineralization than believed earlier. Overall, DOC is strongly affected by the Lena
River plume transporting young Terr-OC from topsoil and/or recently produced vascular pant
material, while near-bottom POC and SOC preferentially carries off-shelf old OC released
from thawing permafrost.
























## 1. Introduction


Studies of terrestrial organic carbon (Terr-OC) in the Arctic Ocean are receiving
increasing interest due to concerns about the consequences on the carbon cycle by amplified
climate change. The Eurasian Arctic Shelf is predicted to experience the highest increase in
temperature on Earth, and its warming is even faster than predicted (Arndt et al., 2015;
Zwiers, 2002). It has been suggested that these changes may translocate increasing amounts
of Terr-OC to the coastal ocean (Vonk and Gustafsson, 2013). The Arctic tundra and taiga
drainage basins represents roughly 50% of the global soil organic matter, much within
shallow permafrost (Gorham, 1991; Tarnocai et al., 2009), and 10–20% of the global
vegetation carbon with about 73% in Eurasia (McGuire et al., 2009; McGuire et al., 2010).
Fluvial and erosional processes are expected to increase, as well as biomass cover, resulting
in higher input fluxes and changing composition of Terr-OC to the continental shelf (Lantuit
et al., 2013; Peterson et al., 2002; Sanchez-Garcia et al., 2014; Serreze et al., 2002). In
addition, those mechanisms would enhance the remobilization of permafrost carbon,
potentially constituting a climate-carbon positive feedback, in terms of $CO_2$ outgassing from
degradation of thawing permafrost. Thus, it is essential to understand the sources, dynamical
fate and composition of exported Terr-OC in order to assess its impact within the carbon
cycle.
The fate of terrestrial OC in the marine system DOC, POC and SOC compartments is
still a matter of debate. Some studies have indicated a conservative behavior of DOC in the
Arctic Ocean with small influence on the ocean-atmosphere exchange of $CO_2$ (Amon and
Meon, 2004; Dittmar and Kattner, 2003a; Köhler et al., 2003; McGuire et al., 2009), little or
no degradation in microbial incubations (Amon and Meon, 2004), and high concentrations of
lignin in the DOC pool (Amon and Benner, 2003; Amon et al., 2012; Lobbes et al., 2000). By
contrast, others suggest that DOC is highly degraded by photochemical oxidation or
microbial respiration in the water column or surface sediments (Alling et al., 2010; Benner
and Kaiser, 2011; Hernes and Benner, 2003; van Dongen et al., 2008b). Investigations of the
particulate compartment indicate that POC degrades much faster than DOC, and just a small
fraction is transported off-shelf within the POC pool (Eglinton and Repeta, 2006; Sanchez-
Garcia et al., 2011; van Dongen et al., 2008b). In addition, other processes on the wide and
shallow Arctic shelves such as hydrodynamic sorting, deposition, resuspension and uptake by
primary production may contribute to the dilution/dispersal of Terr-OC along the water and
sediment dispersal system (Stein and Macdonald, 2004; Tesi et al., 2016; Tesi et al., 2014). It



seems that different pools of Terr-OC have different behavior and fate during remobilization
and transport. DOC and POC pools have much younger $^{14}$C ages than the deposited
sedimentary OC (Guo et al., 2007; Karlsson et al., 2016; Karlsson et al., 2011). Compound-
specific radiocarbon analyses of lipid molecules and lignin phenols of surface sediments and
POC from major river mouths in the Arctic revealed marked age offsets between different
Terr-OC pools (Feng et al., 2013; Vonk et al., 2010). However, we still have a very limited
understanding of the composition and cycling of Terr-OC in the Arctic Ocean.

The East Siberian Arctic Shelf (ESAS) is a particularly relevant region for

investigating the distribution and fate of Terr-OC in the DOC, POC and SOC pools. The
ESAS is the world's largest continental shelf and its adjacent basin is located in a region of
continuous and discontinuous permafrost. The extensive ESAS is quite shallow (~50 m
average depth) and receives massive amounts of Terr-OC. In the west, the Lena river and
coastal erosion are the main inputs of OC (Laptev Sea and western East Siberian Sea, W-
ESS) (Charkin et al., 2011; Salvadó et al., 2015; Semiletov et al., 2011; Tesi et al., 2014;
Vonk et al., 2012). Alternatively, in the eastern East Siberian Sea (E-ESS, from ~160°E to
eastwards) marine phytoplankton represents an important source of OC due to the influence
of nutrient-rich Pacific inflow waters (Semiletov et al., 2005; Stein and Macdonald, 2004).
Many investigations in the Arctic focused on characterizing the composition and fate of
riverine OC (Amon et al., 2012; Benner et al., 2005; Elmquist et al., 2008; Goni et al., 2000;
Lobbes et al., 2000; van Dongen et al., 2008a; Winterfeld et al., 2015), sedimentary OC in the
ESAS (Bröder et al., Submitted; Karlsson et al., 2015; Salvadó et al., 2015; Tesi et al., 2014),
and DOC and POC in the water column of the Eurasian Arctic Shelf (Alling et al., 2010;
Sanchez-Garcia et al., 2011). This is, however, the first study that characterizes collectively
the DOC, POC and SOC pools along the outer shelf. The present study uses carbon isotopes
and macromolecular biomarkers to provide an extensive view of the composition and
distribution of Terr-OC along the outer ESAS, with the objective to evaluate the sources,
degradation and off-shelf transport of the DOC, POC and SOC pools.

**2. Materials and methods**
**2.1 Study area**

The ESAS is the widest, shallowest and, by area, largest continental shelf in the

World Ocean. It comprises 40% of the Arctic shelf and 20% of the Acrtic Ocean (Stein and





Macdonald, 2004). This study focuses on the outer shelf of ESAS seas (Laptev Sea and East
Siberian Sea) and the Kara Sea (Figure 1). The Kara Sea has an area of $880 \cdot 10^3$ km$^2$ and a
mean depth of 110 metres. It receives a large amount of fresh water mainly from the Ob river.
The Laptev Sea, between ~110ºE and 140ºE, covers almost $500 \cdot 10^3$ km$^2$ and has an average
water depth of 50 m. This sea receives large amounts of freshwater (~745 km$^3 \cdot$yr$^{-1}$) mainly
transported by the Lena river (566 km$^3 \cdot$yr$^{-1}$) (Cooper et al., 2008; Semiletov et al., 2000), but
most of the TerrOC that enters the Laptev Sea is coming from coastal erosion of late
Pleistocene ice complex deposits (Semiletov et al., 2011; Vonk et al., 2012). The East
Siberian Sea has an average water depth of 58 m, and is the largest and most ice-bound shelf
sea of the Arctic Ocean (Stein and Macdonald, 2004). It extends from 140ºE to 180ºE
covering an area of $987 \cdot 10^3$ km$^2$, and receives freshwater inputs from the Indigirka and
Kolyma rivers. This sea exhibits two physical and biogeochemical regimes. The eastern East
Siberian Sea (E-ESS, from ~160°E to ~180°E), which is influenced by the Pacific inflow
waters, and primary production represents an important source of OC (Semiletov et al., 2005;
Stein and Macdonald, 2004). And the western East Siberian Sea (W-ESS), between ~140°E
and ~160°E, where river runoff and coastal erosion of thawing permafrost supply the major
part of OC, but there is also a relatively high marine productivity, particularly in certain
polynya regions.

**2.2 Sampling**
A comprehensive set of samples was obtained during July-August 2014 as part of the
international Swedish-Russian-US investigation of the Carbon-Climate-Cryospshere
Interactions in the East Siberian Arctic Ocean (SWERUS-C3) expedition onboard I/B Oden.
The first of two SWERUS-C3 2014 legs was an extensive 45-day campaign of complex
geophysical and hydrogeochemical sampling including at-sea analysis. The sample collection
for this study consisted of four types of samples along the outer ESAS (Figure 1): i) POC
from high volume filtration on 293 mm glass fibre filter (GF/F; Whatman Inc.) with a
nominal 0.7 µm cut-off, ii) POC from 47 mm Teflon filters, iii) DOC isolated with solid
phase extraction (SPE) cartridges, iv) surface sediment samples collected with a multicorer.
Surface and near-bottom waters (5m above bottom) were sampled and filtered through
high volume 293 mm GF/F filters (pre-combusted for 5h at 500ºC). Samples were filtered
either directly from the seawater intake (SWI) or by pumping water from 1000L tanks filled
from the SWI or from a submersible pump. The systems were connected to an electronic flow



meter, in the flow path below the filter, and a pressure meter situated directly above the GF/F
filter holder. We maintained the flow to about 8.5 L·min$^{-1}$, and stopped filtering before the
backpressure reached 1 bar to avoid cell lysing. After sampling of the particulate fraction, the
GF/F filters were folded, put in a pre-combusted aluminium foil and stored at -20°. Since
GF/F filters are not compatible with the alkaline hydrolysis of the CuO oxidation protocol to
analyse lignin-derived phenols, POC samples were also obtained on 47 mm Teflon filters in
order to analyse lignin-derived phenols in POC. We placed the Teflon filters in the filtration
unit and applied a positive pressure flow with a peristaltic pump at a flow rate of 25 ml·min$^{-1}$.
POC samples in Teflon filters were folded in two, placed in petri dishes and stored frozen (-
20°C) until laboratory analysis.
The dissolved fraction of organic matter was isolated by high-volume SPE cartridges
containing 10 g of sorption material composed of octadecyl carbon moieties ($C_{18}$) chemically
bonded to a silica support ($C_{18}$-SPE Mega-Bond Elut; Agilent) (Louchouarn et al., 2000).
Cartridges were preconditioned with 5 resin volumes of methanol followed by 5 resin
volumes of acidified (pH 2) Milli-Q Plus UV water. The water samples, previously filtered
with GF/F, were acidified to pH 2 using reagent-grade concentrated HCl and pumped through
the SPE cartridge with a peristaltic pump and silicone tubing. By this method, the water
(~30L) was delivered directly into the headspace of the SPE cartridge and forced by pressure
through the sorbent at a flow rate of 100 mL min$^{-1}$. Thereafter, we rinsed each SPE cartridge
with 1L of acidified (pH2) Milli-Q Plus UV water to remove residual salts. Sample cartridges
were packed in aluminum foil and stored at 4°C until further processing.
Sediment samples were collected with an 8-tube multicorer (Oktopus GmbH,
Germany), which was developed to collect samples of the seabed with an undisturbed
sediment-water interface. The liners were made of polycarbonate and were 60 cm long with a
10 cm diameter. The multicorer was deployed with full weight (head weight about 500 kg) at
a speed of 0.5 m/s near the seabed. To increase recoveries, the multicorer was left for 1
minute on the seafloor. The cores were sectioned on low resolution (1cm intervals; shelf
stations <200m water depth) or on high resolution (0.5cm intervals; slope and rise stations
>200m water depth), and sediment samples were transferred into plastic bags and stored in
the freezer (-20°C).





### 2.3 Bulk elemental and isotope analysis

The analyses of organic carbon content, $\delta^{13}C$ and $\Delta^{14}C$ in the DOC, POC and SOC pools have been described earlier (Karlsson et al., 2011; Louchouarn et al., 2000). Briefly, DOC was determined onboard after GF/F filtration by high-temperature catalytic oxidation (Shimadzu TOC-L$_{CPH}$). In the laboratory, SPE cartridges were eluted with 50 mL of methanol. Then, we subsampled 0.5-1mL of the eluent, depending on DOC concentrations, and placed it in smooth wall tin capsules for liquids (6 x 12 mm, Elemental Microanalysis, Devon, UK). For organic carbon content and $\delta^{13}C$ composition of POC and SOC, GF/F filters and surface sediment samples were subsampled and acidified with HCl (1.5M) to remove carbonates. The analyses were performed in triplicates using a Carlo Erba NC2500 elemental analyzer connected via a split interface to a Finnigan MAT Delta Plus mass spectrometer at the Stable Isotope Laboratory of the Department of Geological Sciences at Stockholm University. Some subsamples, after similar preparation steps, were analyzed for its radiocarbon content ($\Delta^{14}C$) at the US-NSF National Ocean Sciences Accelerator Mass Spectrometry (NOSAMS) Facility at Woods Hole Oceanographic Institution. Uncertainties of $\Delta^{14}C$, $\delta^{13}C$, and OC analyses were ±0.002 (fraction modern error), ±0.1‰, and ±2% of the measured OC content, respectively.

### 2.4 Lignin phenols analysis

The quantification of lignin-derived phenols in DOC (SPE eluents), POC (Teflon filters), and SOC (surface sediment samples) was performed as described in detail by Louchouarn et al 2000 and Tesi et al., 2014. Briefly, for the analysis of dissolved lignin, 5-15 mL of elution samples (1-2 mg OC equivalent) were reduced to dryness under a stream of nitrogen in Teflon tubes. The dried samples were then oxidized under alkaline oxygen free conditions (degassed NaOH solution, 8%) (Goni and Montgomery, 2000) with an addition of 10 mg of glucose to prevent superoxidation of the lignin polymer and spiked with recovery standards (trans-cinnamic acid and ethyl vanillin). Samples were then acidified, extracted twice with ethyl acetate and concentrated under vacuum at 60ºC. The same oxidation and extraction procedure was also used for POC (Teflon filters) and SOC samples, but without the addition of glucose in sediment samples.

Prior to the analyses, extracts were re-dissolved in pyridine and derivatized. Target compounds were quantified on a gas-chromatograph mass spectrometer (GC-EI-MS, Agilent)





using a DB1-MS capillary column (30m x 250μm, 0.25μm stationary phase thickness,
Agilent J&W) for separation. Quantification of lignin phenols, benzoic acids, and p-
hydroxybenzenes was achieved using the response factors of external standards. All reported
concentrations of CuO oxidation products were reported in mg of biomarker per g OC.

**3. Results and discussion**
**3.1 Elemental Composition and Distribution of DOC, POC and SOC**

DOC in the water column of the outer ESAS during SWERUS-2014 expedition was

one order of magnitude higher than POC. The DOC concentrations ranged from 740 to 3600
$\mu g \cdot L^{-1}$ (mean of $1400 \pm 790 \mu g \cdot L^{-1}$) and POC varied between 20 and 360 $\mu g \cdot L^{-1}$ (mean of
$110 \pm 80 \mu g \cdot L^{-1}$) (Table 1, Figure 2A, Figure 3). Those values are in the same range as previous
studies in the Siberian Arctic Seas (Alling et al., 2010; Benner et al., 2005; Sanchez-Garcia et
al., 2011). Whereas DOC showed the highest values in surface waters of the Laptev Sea
($2000 \pm 1100$ $\mu g \cdot L^{-1}$), particularly off the Lena river mouth, POC concentrations were slightly
higher in the Kara Sea ($290 \pm 86 \mu g \cdot L^{-1}$) and the E-ESS ($150 \pm 92 \mu g \cdot L^{-1}$) with no significant
differences between surface and near-bottom waters. SOC values in surface sediments from
the same stations presented higher concentrations in the E-ESS ($1.32 \pm 0.42\%$), but also
exhibited an increase in the Laptev Sea ($1.21 \pm 0.26\%$) (Figure 3). This is in the lower range of
what was previously reported in the inner-shelf of the ESAS (Charkin et al., 2011; Karlsson
et al., 2015; Karlsson et al., 2011; Tesi et al., 2014; Vonk et al., 2012), suggesting either
degradation of Terr-OC or sediment sorting during the across-shelf transport as discussed in
Tesi et al. 2014 and Bröder et al. 2016. Alternatively, higher POC and SOC values in the E-
ESS may be related to the higher marine productivity in that region due to the Pacific water
influence (Semiletov et al., 2005; Stein and Macdonald, 2004).

The resulting bulk ratios in the $DOC_{SPE}$ fraction indicate terrestrially dominated

organic matter sources. The OC/TN (TN = organic nitrogen + inorganic nitrogen) of $DOC_{SPE}$
ranged between 14 and 43 (mean of $28 \pm 8.4$) without significant differences between surface
and near-bottom waters (Table 1). Those ratios showed decreasing trends off the Lena river
plume with higher ratios in the Laptev Sea and W-ESS. The same pattern and similar ratios
were observed in the inner-shelf of the ESAS (Karlsson et al., 2016). Moreover, these values
are in the same range as OC/TN ratios of DOC in Eurasian Arctic rivers, which varied
between 23 and 69 (Lobbes et al., 2000), and the high OC/TN ratios (>40) of DOC collected





from the Kara Sea (Köhler et al., 2003; Opsahl et al., 1999). Marine organic matter has
OC/TN values around 6-8 and terrestrial derived organic matter OC/TN ratios higher than 15
(Baldock et al., 1992; Hedges et al., 1986; Hedges and Oades, 1997). The OC/TN ratios in
the particulate and sedimentary compartments were much lower than in the $DOC_{SPE}$. Those
ratios ranged between 5 and 12 (mean of 7±1.7) in the POC and from 6 to 8 (mean of 7±0.5)
in the SOC. Similar OC/TN values were observed in the inner-shelf of the ESAS and Arctic
rivers in the particulate fraction (McClelland et al., 2016; Sanchez-Garcia et al., 2011).
However, these lower OC/TN ratios are at odds with e.g. $\delta^{13}$C-OC and may be influenced by
selective degradation of labile carbonaceous forms (Hugelius and Kuhry, 2009), and/or
adsorption of inorganic nitrogen (e.g. ammonium) derived from decomposition of organic
matter (Sanchez-Garcia et al., 2011; Schubert and Calvert, 2001).

**3.2 Stable Carbon and Radiocarbon Isotopes**
The east-to-west extension of the depleted $\delta^{13}$C signatures reflects a strong influence
of the Lena River (Figure 3), both in the Laptev Sea and the ESS. The $\delta^{13}$C signatures in the
three carbon pools ranged from -23.9±1.9‰ in the SOC, -26.1±1.2‰ in the $DOC_{SPE}$ and -
27.1±1.9‰ in the POC, with no significant differences between surface and bottom waters
(Tables 1, 2 and 3). The more depleted $\delta^{13}$C-POC is consistent with marine productivity
using excess dissolved inorganic carbon (DIC) from the Lena river, which is more depleted
than marine DIC (Alling et al., 2012; Semiletov et al., 2016). This mechanism also explains
similarly depleted $\delta^{13}$C-POC in the Lena plume far offshore in the Laptev Sea and ESS that
matched with depletion of other nutrients (Alling et al., 2010; Sanchez-Garcia et al., 2011).
The distribution of $\delta^{13}$C-SOC was more homogeneous reflecting average over time in the
surface sediment regime. Only the concentration of DOC presented a good correlation with
$\delta^{13}$C-$DOC_{SPE}$ signatures, which indicates that higher concentrations of DOC come from
terrigenous sources (Figure 3).
The radiocarbon ages of $DOC_{SPE}$ and POC showed a depleted and younger trend off
the Lena River plume. The $\Delta^{14}$C signals ranged between -395±83‰ (SOC), -226±92‰
($DOC_{SPE}$) and -113±122‰ (POC) presenting contrasting offsets between the Laptev Sea and
the East Siberian Sea, particularly in the E-ESS (Tables 1, 2 and 3; Figure 4). The older and
enriched $\delta^{13}$C signatures in the outer-shelf of the ESS suggest the influence of sea ice
coverage and the Pacific inflow from the East. The more enriched $\Delta^{14}$C signatures in POC
than in DOC are in accordance with previous studies in the Arctic Ocean (Griffith et al.,




2012) and the Southern Ocean (Druffel and Bauer, 2000), which reflect a dominant marine
source in the particulate carbon pool. The SOC pool does not present such marked west-east
distribution of $\Delta^{14}C$ as observed in $DOC_{SPE}$ and POC. The SOC also depicts older signatures
near the New Siberian Islands. A recent study from the same area at the land-ocean interface
presented older signatures in the POC than in the DOC (Karlsson et al., 2016), suggesting
that thawing permafrost was transported preferentially within the POC pool. Therefore, our
results support the hypothesis that remobilized permafrost preferentially settles out close to
land, and then it is transported off-shelf through sediment resuspension-redeposition events.
The older signals in the dissolved fraction of the ice-covered regions are consistent with a
more recalcitrant OC in the dissolved pool of the Arctic Ocean (Follett et al., 2014; Griffith et
al., 2012). It seems that the ice extent boundary works as a barrier that prevents the input of
young DOC coming from the buoyant freshwater plume of the Lena river (Figure 4).  It is
important to point out that near-bottom waters presented more depleted and similar $\Delta^{14}C$
signatures in both $DOC_{SPE}$ and POC (-258±94‰ and -250±83‰, respectively) than in surface
waters (-213±93‰ and -57±86‰, respectively) (Figure 5; Tables 1 and 2). Those contrasting
age offsets between surface and near-bottom waters, particularly for the POC fraction, may
reflect the off-shelf transport of OC translocated over long distances from thawing
permafrost.
DOC was the only carbon pool that presented good correlations with $\Delta^{14}C$ and $\delta^{13}C$
data (Figure 6). Those relationships are consistent with previous observations in the Arctic
Ocean (Amon et al., 2012; Benner et al., 2004; Schreiner et al., 2013). The correlation
between $\delta^{13}C$ and DOC ($r^2$=0.68) reflects the processes of the terrigenous DOC along the off-
shelf transport, with more depleted signatures in higher DOC samples and enriched
signatures in lower DOC concentrations. Processes such as hydrodynamic sorting, deposition,
resuspension and uptake by primary production may contribute to the dispersal and
processing of the OC in the ESAS. On the other hand, the relationship between $\Delta^{14}C$ and
DOC ($r^2$=0.87) represents the source of the terrigenous DOC, where higher DOC samples are
composed by young Terr-OC and lower DOC concentrations by old and refractory Terr-OC.
Overall, these findings are direct evidence that a large proportion of DOC exported to the
outer shelf comes from young and fresh vascular plant material.




### 3.3 Lignin-Derived Phenols

Lignin-derived phenols are exclusively synthesized by vascular plants and account for one third of the organic matrix of wood, grasses, needles, and herbage, therefore, they have been extensively used to characterize the pathway of terrestrial matter in the marine environment (Louchouarn et al., 1999; Pasqual et al., 2013; Tesi et al., 2014). The carbon-normalized lignin content (mg/g OC) refers to the sum of vanillyl, syringyl and cinnamyl phenols. $DOC_{SPE}$ samples presented lignin concentration on the same order as the corresponding underlying sediments (0.10-2.34 and 0.08-2.40 mg/g OC, respectively). By contrast, the particulate carbon pool had slightly lower OC-normalized lignin concentrations between 0.03 and 1.14 mg/g OC (Figure 4; Tables 1 and 2). Lignin levels in the SOC and DOC pools are in agreement with previous studies in ESAS sediments (Karlsson et al., 2015; Tesi et al., 2014) and the polar surface water of the Arctic Ocean (Benner et al., 2005). These are the first POC-lignin data in the Arctic Ocean. Lignin concentrations exhibited contrasting offsets in surface and near-bottom waters depending on the carbon pool. $DOC_{SPE}$ presented similar lignin concentrations in surface and near-bottom waters, except for the more concentrated samples closer to the Lena river mouth. Conversely, POC showed enhanced levels in all near-bottom water samples (from 0.17 to 1.14 mg/g OC), with even higher concentrations than in the dissolved pool (from 0.16 to 0.91 mg/g OC) (Figure 2B; Tables 1 and 2). Those vertical lignin dissimilarities in the water column were not observed in the total OC of the dissolved and particulate fractions. However, $\Delta^{14}C$-OC also showed offsets in the particulate pool. While $DOC_{SPE}$ depicted similar $\Delta^{14}C$ signatures in both surface and near-bottom waters, POC was much older in near-bottom waters. Hence, these findings suggest that particulate old OC with high concentrations of lignin, probably coming from thawing permafrost, is mainly transported off-shelf in near-bottom waters by resuspension and remobilization of the SOC pool.

Lignin phenols exhibited decreasing OC-normalized concentrations with increasing distance from the Lena river plume in all carbon pools and in both surface and near-bottom waters (Figure 4). Previous studies in ESAS for other biomarkers have also reported decreasing across-shelf trends of terrestrial organic matter with increasing distance from the coast (Selver et al., 2015; Tesi et al., 2014). Several studies reported minimal degradation of DOM across the broad Eurasian shelves (Dittmar and Kattner, 2003b; Kattner et al., 1999; Köhler et al., 2003). With such a scenario, our off-shelf decreasing lignin concentrations in DOC, POC and SOC pools may be interpreted to result from dilution with marine organic



matter during transport and/or hydrodynamic sorting along the water and sediment dispersal
system. However, other studies found that terrestrial DOC in this ESAS shelf sea system was
degraded, with a first-order removal rate constant of 0.3 $yr^{-1}$ (Alling et al., 2010). Recent
studies also suggested high reactivity of lignin in rivers (Benner and Kaiser, 2011; Fichot and
Benner, 2014; Ward et al., 2013) and in offshoreward direction across ESAS (Bröder et al.,
Submitted; Tesi et al., 2014). If this instead is the dominating process, the decreasing trend in
the current study may also be due to degradation.

Our results depicted a strong positive relationship between lignin phenols and total

dissolved organic content within the 35 $DOC_{SPE}$ samples analysed along the outer ESAS (r =
0.89) (Figure 7). There were also significant correlations between OC-normalized
concentrations of lignin phenols and $\delta^{13}C$ (r = 0.66) and $\Delta^{14}C$ (r = 0.78) in the $DOC_{SPE}$ pool.
These data is consistent with the modern radiocarbon ages of DOC observed in Arctic rivers
(Benner et al., 2004; Benner et al., 2005; Karlsson et al., 2016), which also demonstrated a
general agreement between lignin phenols and $\Delta^{14}C$ signatures as traces of terrrigenous DOC.
Lignin phenols were found in old OC from permafrost (Tesi et al., 2014). Compound-specific
radiocarbon analyses of lignin phenols from sediments off major river mouths in ESAS
indicated that those macromolecules were younger than sedimentary bulk OC (Feng et al.,
2013). This is consistent with lignin compounds derived from both sources, and the higher
lignin content from younger $DOC_{SPE}$ likely coming from either recently produced vascular
plant material or from contemporary topsoil. By contrast, the slightly negative correlation
between lignin phenols and $\Delta^{14}C$ signatures in POC (r = 0.53) (Figure 7), with higher lignin
concentrations in older POC, suggests that those macromolecules are coming from
remobilized older permafrost carbon. Those results are consistent with previous findings
indicating that OC from thawed permafrost is transported preferentially within the particulate
carbon pool (Karlsson et al., 2016). There was no relationship between lignin content and
bulk POC and SOC, which suggests that both pools are composed by a mixture of marine and
terrestrial organic carbon. Taken together, whereas young Terr-OC is transported mainly
within the dissolved fraction, near-bottom POC and SOC carries off-shelf preferentially old
OC from remobilized permafrost.





### 3.4 Biomarker indications of sources of DOC, POC and SOC

The ratios of individual or classes of lignin phenols are frequently used to infer the types of plants yielding the phenols and to what extent the organic matter has been oxidized. Vanillyl phenols (vanillin, acetovanillone and vanillic acid) are ubiquitous in lignin, while syringyl phenols (syringaldehyde, acetosyringone and syringic acid) derive only from angiosperms (Hedges and Mann, 1979; Spencer et al., 2008). Ratios of syringyl to vanillyl (S/V) phenols indicate contribution of angiosperm and gymnosperm vegetation to Terr-OC. Our low S/V ratios (from 0.14 to 1.05) indicate gymnosperm vegetation as the most important source of lignin (Figure 8A; Tables 1, 2 and 3). However, the high S/V ratios in the easternmost samples, particularly within the DOC and SOC, reflect a higher source apportion of tundra plants (Lobbes et al., 2000). Elevated values of S/V were also reported in sediments and dissolved organic carbon from the inner-shelf of the same study area (Karlsson et al., 2016; Tesi et al., 2014). The fact that the Indigirka and Kolyma watersheds are north of the Arctic Cirle with a general shift to flowering tundra plants could explain the elevated S/V ratios in the E-ESS.

Cinnamyl phenols (*p*-coumaric acid, ferulic acid) are predominantly found in herbaceous tissues, and the ratio cinnamyl over vanillyl (C/V) has been used to distinguish woody lignin from other sources (Goni and Hedges, 1992; Hedges and Mann, 1979). C/V ratios did not show a specific trend along the east to west data set. Similar results were observed previously in inner-shelf sediments and in the colloidal DOC fraction from the ESAS (Karlsson et al., 2016). As we also analysed lignin phenols in the particulate fraction, we could see that C/V ratios were slightly higher in POC ($0.64\pm0.42$) than in SOC ($0.37\pm0.18$) (Tables 1, 2 and 3), possibly reflecting more herbaceous plants or sphagnum moss source in the particulate pool and more woody lignin in the sedimentary carbon. However, those ratios should always be carefully interpreted as photooxidation and microbial degradation can alter the original compositions (Hedges and Prahl, 1993; Opsahl and Benner, 1995). Regarding the classical source plot of S/V versus C/V, our data set distributes along a line between angiosperm leaves and grasses and gymnosperm wood, suggesting that little amounts of non-woody angiosperm tissues are mixing with large amounts of gymnosperm woods in these samples (Figure 8A). Overall, this plot underlines distinct clustering between OC pools, suggesting that angiosperms are mainly transported by SOC and gymnosperms by POC.



*p*-hydroxybenzoic acids (P) can originate from different sources, while *p*-
hydroxyacetophenone (Pn) has only been detected in terrigenous organic matter, particularly
in peat and sphagnum (Williams et al., 1998), while *p*-hydroxybenzaldehyde (Pl) and *p*-
hydroxybenzoic acid (Pd) can also derive from marine sources (Goni and Hedges, 1995). The
Pn/P ratios observed in DOC (0.08-0.37), SOC (0.06-0.17) and POC (0.02-0.14) indicate that
the OC in the dissolved pool is more terrestrial than in the particulate and sediment pools
(Tables 1, 2 and 3; Figure 9A). Those ratios present a slight east-to-west trend with higher
values off the Lena river plume. Similar trends and results were observed by Karlsson et al.,
2016 in the colloidal OC along the ESAS coast (0.15-30). Amon et al., 2012 characterized the
chemical composition of DOC in Arctic rivers and reported Pn/P ratios in the same range, for
instance, those ratios in the Lena, Indigirka and Kolyma rivers varied between 0.30 and 0.39.
It is important to point out that these findings are in agreement with the relationships of DOC
and $\Delta^{14}C$ and $\delta^{13}C$ presented above, which indicate that the DOC exported off-shelf is mainly
"young" and terrestrial. By contrast, P/V ratios presented an opposite trend with much higher
values in the POC than in the other carbon pools (Tables, 1, 2, 3). This distribution indicates
that POC is mainly composed by marine OC with enhanced concentrations in the E-ESS
reflecting the Pacific inflow from the East. Overall, these proxies

### 3.5 Indicators of Terr-OC degradation across the OC continuum

The relative abundances of some lignin phenols provide information about the
diagenetic alteration of Terr-OC. The acid/aldehyde ratios of syringyl (Sd/Sl) and vanillyl
(Vd/Vl) have been utilized as indicators of the relative degradation of the plant matter
contribution, as aldehydes degrade faster than corresponding acids (Goni and Hedges, 1992;
Hedges et al., 1986). However, some caution should be applied in the interpretation as source
signals are more varied than originally thought, and fractionation occurs during
leaching/adsorption processes (Benner et al., 1990; Hernes et al., 2007). Our data showed
Sd/Sl and Vd/Vl ratios higher in $DOC_{SPE}$ (1.9±0.6 and 2.0±0.7, respectively), than in SOC
(0.9±0.2 and 1.1±0.3) and POC (0.4±0.1 and 0.5±0.2) indicating the presence of highly
oxidized lignin in the dissolved pool (Tables 1, 2 and 3; Figure 8B). It is important to notice
that the ranges of Sd/Sl and Vd/Vl ratios in this study were relatively broad and with clear
clusters between carbon pools. POC ratios were lower than the underlying sediments and the
dissolved carbon pool presented very high ratios (Figure 8B). Those ratios are in accordance
with previous studies in POC (Hernes and Benner, 2002; Lobbes et al., 2000; Winterfeld et



al., 2015), sediments (Goni and Montgomery, 2000; Goni et al., 2005; Tesi et al., 2014) and
DOC (Amon et al., 2012; Hernes and Benner, 2002; Lobbes et al., 2000), which also found
higher ratios in the dissolved than in the particulate phase. The elevated Sd/Sl and Vd/Vl in
the dissolved fraction, as well as the enhanced ratios in the SOC pool, may reflect
leaching/adsorption processes (Hernes et al., 2007; Houel et al., 2006).

Another proxy commonly used to determine the degradation of Terr-OC is the ratio

between 3,5-dihydroxybenzoic acid and vanillyl phenols (3,5-Bd/V) (Farella et al., 2001;
Houel et al., 2006; Otto and Simpson, 2006; Prahl et al., 1994). Since 3,5-Bd is highly
resistant to degradation (Dickens et al., 2007) while vanillyl phenols are very susceptible to
degradation, higher values of 3,5-Bd/V are indicative of more degraded Terr-OC. Our results
presented opposite patterns than the ones observed by Sd/Sl and Vd/Vl with higher 3,5-Bd/V
ratios in POC (1.7±0.7) and SOC (1.0±0.6) and lower ratios in DOC$_{SPE}$ (0.7±0.3) (Tables 1, 2
and 3; Figure 9B). These values are in accordance with those in DOC from ESAS rivers (0.4-
0.7) (Amon et al., 2012) and the colloidal fraction from the ESAS land-ocean interface (0.4-
0.8) (Karlsson et al., 2016). In addition, SOC ratios are consistent with those observed in
surficial sediments from the same area (0.2-1.3) (Tesi et al., 2014). However, we could not
find previous studies to compare our 3,5-Bd/V ratios in POC. The higher ratios in POC
suggest that Terr-OC is more degraded in the particulate fraction than in the other carbon
pools of the outer ESAS. We also should consider that macroalgal sources of 3,5-Bd might be
significant in selected marine systems comprising minimal fractions of terrigenous organic
matter (Goni and Hedges, 1995). The 3,5-Bd/V ratios in DOC$_{SPE}$ and POC depicted a slightly
increasing tendency in the eastern samples (Figure 9B). Previous studies in sediments and the
colloidal fraction from the ESAS also reported the same trend (Karlsson et al., 2016; Tesi et
al., 2014) reflecting the Pacific inflow from the east of more marine and/or degraded OC. We
consider in our study that this degradation proxy is more reliable than Sd/Sl and Vd/Vl ratios
as it is not affected by the leaching/adsorption processes between carbon pools. Therefore,
the Terr-OC in the ESAS is more degraded in the POC and SOC pools.

Two cinnamyl phenols, *p*-coumaric acid (pCd) and ferulic acid (Fd), are additional

CuO oxidation products of lignin that are particularly abundant in grasses and many
herbaceous tissues. The two phenols differ by a presence of a methoxyl group, and this may
explain the preferential degradation of ferulic acid (Opsahl and Benner, 1998). Therefore,
pCd/Fd ratio has been used as a diagenetic indicator (Amon et al., 2012; Houel et al., 2006).
In this data set pCd/Fd ratios follow the same pattern as the ones observed in 3,5-Bd/V ratios



with higher values in POC and SOC (Tables 1, 2 and 3; Figure 9C). This strengthens the
hypothesis that POC and SOC are more degraded than DOC.
The strong relationship between lignin concentrations and the $^{14}$C-age of DOC$_{SPE}$ also
reflects the role of diagenetic processes. The younger the marine DOC is, the higher is the
concentration of lignin (Figure 7). Those relationships are consistent with previous
observations in the Arctic Ocean where the age of DOC decreased with increasing
concentration of lignin (Benner et al., 2004). These results suggest that a large proportion of
DOC exported to the outer shelf of the ESAS, off the Lena river, comes from recently
produced vascular plant material with little exposure to microbial degradation. Whereas most
of terrigenous POC settles out close to land and is transported through repeated cycles of
deposition and resuspension across the shelf, DOC is dispersed further out onto the EAS with
variable extends of conservative mixing.

**4. Conclusions**
This extensive study provides improved understanding on the sources and
composition of Terr-OC in the DOC, POC and SOC pools in the extensive outer ESAS. The
distribution of a wide variety of bulk ($\delta^{13}$C and $\Delta^{14}$C) and macromolecular proxies (lignin-
derived phenols) reflects a strong influence of the Lena river on the outer shelf, both in the
Laptev Sea and the western ESS. These findings demonstrate that a large proportion of the
DOC exported off-shelf comes from "young" and fresh vascular plant material. The older and
more enriched $\delta^{13}$C signatures in the E-ESS and its higher POC and SOC concentrations
suggest a greater influence of sea ice coverage and the Pacific inflow. The geochemical
proxies show dominance of marine sources in the POC pool. However, near-bottom waters
present more depleted $\Delta^{14}$C signatures and higher concentrations of lignin, particularly for the
POC fraction. This may reflect the off-shelf transport of permafrost-derived OC in the
nepheloid layer, through repeated cycles of deposition and resuspension across the shelf. The
ratios of S/V indicate gymnosperm vegetation as the most important source of lignin, and
increasing S/V ratios in the easternmost samples reflect a relatively higher source
contribution of tundra plants. Moreover, the opposite trends in the Pn/P and P/V ratios
confirm the terrigenous source of DOC and the marine composition of POC, particularly in
the E-ESS. Taking together S/V and C/V ratios we observe distinct clustering between DOC,
POC and SOC, suggesting that those pools are carrying Terr-OC of partially different origin.
Regarding the degradation state of Terr-OC, lignin-phenols fingerprints are presenting





contrasting results. While acid/aldehyde ratios are higher for DOC, possibly due to
fractionation during leaching, 3,5-Bd/V and pCd/Fd ratios were enhanced in POC and SOC,
suggesting degradation. If this hypothesis is true, the remobilized OC from permafrost, which
is mainly transported within these carbon pools, could experience less burial and more
mineralization than believed earlier. Overall, the high abundance of Terr-OC in the outer
ESAS, particularly in the dissolved and sedimentary carbon pools, is a clear indicator of the
magnitude of shelf to basin transport. Taken together, the results suggest that DOC, POC and
SOC are composed of partially different Terr-OC. While DOC is strongly affected by
buoyant freshwater plumes transporting young Terr-OC from topsoil and/or recently
produced vascular pant material, near-bottom POC and SOC carries off-shelf old OC released
from thawing permafrost.

**Acknowledgements**
We thank the crew and personnel of the international SWERUS-C3 expedition 2014
onboard *I/B Oden* and the Swedish Polar Research Secretariat for logistics support. This
study was supported by the Knut and Alice Wallenberg Foundation, Headquarters of the
Russian Academy of Sciences (RAS), and the Far Eastern Branch of the RAS, the Swedish
Research Council (VR contract 621-2004-4039 and 621-2007-4631), the Russian Foundation
of Basic Research (08-05-13572, 08-05-00191-a, and 07-05-00050a), the Nordic Council of
Ministers Cryosphere-Climate-Carbon Initiative (project Defrost, contract 23001), the
European Research Council (ERC-AdG project CC-TOP #695331) and the US National
Science Foundation (OPP ARC 0909546). Contribution of T. Tesi is 1876 of ISMAR-CNR
Sede di Bologna. I.P. Semiletov thanks the Russian Government for support (megagrant
#2013–220–04–157 under contract 14.Z50.31.0012).    J.A. Salvadó acknowledges EU
financial support as a Marie Curie grant (FP7-PEOPLE-2012-IEF; project 328049).








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





**Table 1. Composition of surface and near-bottom DOC samples collected in the outer Eurasian**
**Arctic Shelf.**

| ID | Region | Lat | Long | DOC[1] | OC/TN | $\delta^{13}C$ | $\Delta^{14}C$ | Lignin[2] | S/V | C/V | Sd/Sl | Vd/Vl | 3.5Bd/V |
|---|---|---|---|---|---|---|---|---|---|---|---|---|---|
| *DOC-swi* | | | | | | | | | | | | | |
| T-1 | KS | 79.8 | 67.9 | 880 | 14 | -24.7 | | 0.08 | 0.45 | 0.51 | 2.62 | 3.12 | 1.24 |
| T-2 | KS | 81.7 | 75.3 | 910 | 15 | -24.6 | | 0.11 | 0.45 | 0.59 | 1.83 | 2.12 | 1.28 |
| T-3 | LS | 81.3 | 109.4 | 810 | 18 | -25.1 | | 0.29 | 0.46 | 0.33 | 1.64 | 1.88 | 0.81 |
| T-4 | LS | 81.0 | 112.9 | 1200 | 18 | -25.4 | | 0.19 | 0.43 | 0.32 | 2.04 | 2.40 | 0.85 |
| 1 | LS | 78.9 | 125.2 | 740 | 18 | -24.9 | -279 | 0.25 | 0.47 | 0.43 | 1.67 | 1.92 | 0.96 |
| 4 | LS | 77.8 | 126.7 | 1200 | 19 | -25.9 | -214 | 0.45 | 0.41 | 0.21 | 2.65 | 2.70 | 0.58 |
| 6 | LS | 77.1 | 127.4 | 1100 | 18 | -26.0 | | | | | | | |
| 13 | LS | 76.8 | 125.9 | 1300 | 22 | -26.3 | | | | | | | |
| 14 | LS | 76.9 | 127.8 | 1400 | 25 | -26.8 | | 0.51 | 0.37 | 0.19 | 1.75 | 1.99 | 0.41 |
| 23 | LS | 76.2 | 129.3 | 3400 | 43 | -27.8 | | 2.40 | 0.32 | 0.20 | 1.49 | 1.53 | 0.24 |
| 24 | LS | 75.6 | 129.6 | 3300 | 42 | -27.9 | | 2.05 | 0.31 | 0.17 | 1.62 | 1.49 | 0.29 |
| 25 | LS | 76.0 | 130.7 | 3600 | 43 | -27.9 | -19 | 2.01 | 0.31 | 0.15 | 1.85 | 1.71 | 0.26 |
| 26 | LS | 76.5 | 132.0 | 3400 | 40 | -27.6 | | | | | | | |
| 27 | LS | 76.9 | 132.2 | 2200 | 36 | -27.4 | | | | | | | |
| 28 | LS | 77.3 | 134.8 | 2650 | 39 | -27.6 | -90 | 1.22 | 0.33 | 0.17 | 1.64 | 1.69 | 0.33 |
| 29 | LS | 77.8 | 136.7 | 1700 | 33 | -27.0 | | | | | | | |
| 39 | W-ESS | 77.7 | 141.4 | 2000 | 36 | -27.4 | | 0.80 | 0.34 | 0.23 | 1.69 | 1.38 | 0.46 |
| 40 | W-ESS | 77.6 | 145.8 | 2200 | 36 | -27.2 | | 0.70 | 0.36 | 0.25 | 1.99 | 1.78 | 0.34 |
| 41 | W-ESS | 77.0 | 148.3 | 1400 | 35 | -29.0 | | | | | | | |
| 44 | W-ESS | 76.3 | 146.0 | 1300 | 26 | -26.4 | -160 | 0.41 | 0.43 | 0.33 | 1.93 | 1.74 | 0.51 |
| 45 | W-ESS | 76.4 | 148.1 | 1100 | 30 | -26.9 | | 0.31 | 0.48 | 0.43 | 2.81 | 2.99 | 0.71 |
| 46 | W-ESS | 76.4 | 149.9 | 935 | 25 | -26.3 | | | | | | | |
| 48 | W-ESS | 76.5 | 150.8 | 995 | 22 | -26.2 | | 0.22 | 0.47 | 0.46 | 2.93 | 3.06 | 0.97 |
| 49 | W-ESS | 76.5 | 156.9 | 1300 | 23 | -25.4 | | 0.13 | 0.59 | 0.51 | 2.81 | 3.55 | 0.90 |
| 50 | W-ESS | 75.8 | 158.5 | 1100 | 21 | -25.3 | -262 | 0.16 | 0.55 | 0.49 | 3.09 | 3.07 | 0.98 |
| 52 | E-ESS | 74.1 | 160.6 | 880 | 24 | -25.3 | -288 | 0.13 | 0.38 | 0.14 | 0.65 | 1.01 | 0.61 |
| 56 | E-ESS | 74.6 | 161.9 | 920 | 21 | -24.9 | | | | | | | |
| 57 | E-ESS | 74.4 | 163.7 | 850 | 23 | -24.5 | | 0.20 | 0.66 | 0.86 | 1.26 | 1.76 | 0.88 |
| 58 | E-ESS | 74.4 | 166.2 | 850 | 23 | -24.7 | | 0.15 | 0.54 | 0.49 | 3.47 | 3.77 | 1.26 |
| 59 | E-ESS | 74.4 | 168.5 | 750 | 25 | -25.0 | | | | | | | |
| 60 | E-ESS | 73.4 | 169.5 | 850 | 21 | -24.6 | -268 | 0.23 | 0.68 | 0.70 | 1.57 | 1.91 | 0.64 |
| 61 | E-ESS | 74.1 | 170.9 | 890 | 26 | -24.4 | -278 | 0.20 | 0.64 | 0.80 | 2.28 | 2.81 | 1.48 |
| 63 | E-ESS | 74.7 | 172.4 | 820 | 27 | -25.3 | | 0.22 | 0.59 | 0.78 | 1.94 | 2.36 | 0.75 |
| 66 | E-ESS | 75.9 | 174.3 | 860 | 26 | -26.0 | -270 | 0.19 | 0.56 | 0.88 | 1.72 | 2.26 | 0.94 |
| *DOC-sub* | | | | | | | | | | | | | |
| 13 | LS | 76.8 | 125.9 | 1200 | 25 | -26.7 | | | | | | | |
| 14 | LS | 76.9 | 127.8 | 1200 | 26 | -26.5 | | 0.24 | 0.52 | 0.55 | 2.50 | 2.54 | 0.70 |
| 23 | LS | 76.2 | 129.3 | 1200 | 30 | -27.0 | | | | | | | |
| 25 | LS | 76.0 | 130.7 | 1100 | 35 | -27.2 | | 0.91 | 0.40 | 0.35 | 1.39 | 1.21 | 0.35 |
| 27 | LS | 76.9 | 132.2 | 1100 | 30 | -26.3 | | | | | | | |
| 28 | LS | 77.3 | 134.8 | 1700 | 36 | -27.4 | -171 | 0.86 | 0.37 | 0.33 | 1.17 | 0.95 | 0.31 |
| 29 | LS | 77.8 | 136.7 | 920 | 33 | -26.6 | | | | | | | |
| 39 | W-ESS | 77.7 | 141.4 | 2200 | 46 | -28.6 | | | | | | | |
| 40 | W-ESS | 77.6 | 145.8 | 2000 | 36 | -27.6 | | 0.61 | 0.39 | 0.38 | 1.74 | 1.40 | 0.41 |
| 41 | W-ESS | 77.0 | 148.3 | 1500 | 42 | -28.0 | | | | | | | |
| 44 | W-ESS | 76.3 | 146.0 | 1300 | 32 | -27.4 | -188 | 0.28 | 0.55 | 0.67 | 1.21 | 1.28 | 0.47 |
| 46 | W-ESS | 76.4 | 149.9 | 1100 | 19 | -25.7 | | | | | | | |
| 48 | W-ESS | 76.5 | 150.8 | 990 | 26 | -26.4 | | 0.16 | 0.57 | 0.73 | 2.11 | 2.09 | 0.65 |
| 50 | W-ESS | 75.8 | 158.5 | 970 | 20 | -26.9 | | 0.20 | 0.58 | 0.69 | 1.41 | 1.73 | 0.63 |
| 52 | E-ESS | 74.1 | 160.6 | 940 | 20 | -25.3 | -307 | 0.19 | 0.66 | 0.74 | 1.18 | 1.44 | 0.71 |
| 56 | E-ESS | 74.6 | 161.9 | 910 | 21 | -25.4 | | | | | | | |
| 59 | E-ESS | 74.4 | 168.5 | 720 | 20 | -24.9 | | | | | | | |
| 60 | E-ESS | 73.4 | 169.5 | 870 | 20 | -25.4 | -366 | 0.26 | 0.68 | 0.72 | 1.16 | 1.45 | 0.57 |
| 63 | E-ESS | 74.7 | 172.4 | 820 | 18 | -24.6 | | 0.19 | 0.71 | 0.75 | 1.13 | 1.44 | 0.65 |

[1]DOC concentrations ($\mu g \cdot L^{-1}$)
[2]Lignin OC-normalized concentrations ($mg \cdot g^{-1}$ OC)

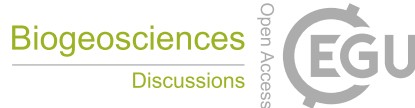



**Table 2. Composition of surface and near-bottom POC samples collected in the outer Eurasian**
**Arctic Shelf.**

| ID | Region | Lat | Long | POC[1] | OC/TN | $\delta^{13}$C | $\Delta^{14}$C | Lignin[2] | S/V | C/V | Sd/Sl | Vd/Vl | 3.5Bd/V |
|---|---|---|---|---|---|---|---|---|---|---|---|---|---|
| *POC-swi* | | | | | | | | | | | | | |
| T-1 | KS | 79.8 | 67.9 | 230 | 5.1 | -26.9 | | 0.16 | 0.37 | 0.70 | 0.29 | 0.40 | 1.93 |
| T-2 | KS | 81.7 | 75.3 | 350 | 6.0 | -24.7 | | 0.21 | 0.42 | 1.58 | 0.35 | 0.59 | 2.72 |
| T-3 | LS | 81.3 | 109.4 | 85 | 6.5 | -28.2 | | 0.71 | 0.14 | 0.25 | 0.33 | 0.27 | 0.95 |
| T-4 | LS | 81.0 | 112.9 | 89 | 5.9 | -28.0 | | 0.58 | 0.31 | 0.77 | 0.27 | 0.45 | 2.26 |
| 1 | LS | 78.9 | 125.2 | 70 | 4.9 | -28.6 | 14 | 0.13 | 0.31 | 0.50 | 0.45 | 0.42 | 1.45 |
| 4 | LS | 77.8 | 126.7 | 140 | 6.3 | -28.8 | 11 | 0.14 | 0.32 | 0.65 | 0.48 | 0.57 | 3.05 |
| 6 | LS | 77.1 | 127.4 | 100 | 5.4 | -26.8 | | | | | | | |
| 13 | LS | 76.8 | 125.9 | 51 | 5.4 | -27.5 | | | | | | | |
| 14 | LS | 76.9 | 127.8 | 54 | 4.9 | -27.9 | | 0.44 | 0.30 | 0.38 | 0.32 | 0.33 | 1.62 |
| 23 | LS | 76.2 | 129.3 | 40 | 5.1 | -29.0 | | 0.72 | 0.46 | 0.44 | 0.78 | 1.09 | 0.72 |
| 24 | LS | 75.6 | 129.6 | 67 | 5.4 | -28.4 | | 0.57 | 0.44 | 0.21 | 0.64 | 0.97 | 0.38 |
| 25 | LS | 76.0 | 130.7 | 120 | 5.4 | -30.2 | 5 | 0.17 | 0.43 | 0.29 | 0.60 | 0.85 | 0.66 |
| 26 | LS | 76.5 | 132.0 | 78 | 5.0 | -29.5 | | | | | | | |
| 27 | LS | 76.9 | 132.2 | 96 | 5.1 | -29.6 | | | | | | | |
| 28 | LS | 77.3 | 134.8 | 45 | 5.2 | -29.3 | -75 | 0.24 | 0.41 | 0.42 | 0.42 | 0.57 | 0.82 |
| 29 | LS | 77.8 | 136.7 | 58 | 5.1 | -28.7 | | | | | | | |
| 39 | W-ESS | 77.7 | 141.4 | 69 | 8.1 | -28.0 | | 0.21 | 0.52 | 0.80 | 0.24 | 0.43 | 1.77 |
| 40 | W-ESS | 77.6 | 145.8 | 85 | 8.6 | -28.5 | | 0.12 | 0.32 | 0.39 | 0.39 | 0.44 | 1.24 |
| 41 | W-ESS | 77.0 | 148.3 | 66 | 7.0 | -27.8 | | | | | | | |
| 44 | W-ESS | 76.3 | 146.0 | 63 | 7.9 | -28.8 | 64 | 0.11 | 0.52 | 0.34 | 0.23 | 0.34 | 1.21 |
| 45 | W-ESS | 76.4 | 148.1 | 61 | 8.4 | -27.5 | | | | | | | |
| 46 | W-ESS | 76.4 | 149.9 | 27 | 6.7 | -27.5 | | | | | | | |
| 48 | W-ESS | 76.5 | 150.8 | 52 | 7.0 | -27.3 | | 0.06 | 0.33 | 0.59 | 0.39 | 0.63 | 1.61 |
| 49 | W-ESS | 76.5 | 156.9 | 140 | 9.1 | -25.9 | | 0.05 | 0.35 | 0.70 | 0.40 | 0.56 | 1.82 |
| 50 | W-ESS | 75.8 | 158.5 | 96 | 9.3 | -25.6 | -102 | 0.05 | 0.32 | 0.63 | 0.42 | 0.48 | 1.30 |
| 52 | E-ESS | 74.1 | 160.6 | 120 | 8.1 | -24.4 | -94 | 0.06 | 0.58 | 0.96 | 0.31 | 0.47 | 2.43 |
| 56 | E-ESS | 74.6 | 161.9 | 150 | 9.2 | -24.0 | | | | | | | |
| 57 | E-ESS | 74.4 | 163.7 | 160 | 10.9 | -23.0 | | 0.05 | 0.32 | 0.73 | 0.31 | 0.52 | 2.11 |
| 58 | E-ESS | 74.4 | 166.2 | 60 | 7.3 | -23.7 | | 0.08 | 0.43 | 0.79 | 0.38 | 0.74 | 2.52 |
| 59 | E-ESS | 74.4 | 168.5 | 290 | 10.2 | -27.4 | | | | | | | |
| 60 | E-ESS | 73.4 | 169.5 | 110 | 5.9 | -25.4 | -90 | 0.05 | 0.44 | 0.87 | 0.47 | 0.45 | 1.56 |
| 61 | E-ESS | 74.1 | 170.9 | 230 | 8.0 | -24.9 | -69 | 0.03 | 0.72 | 2.39 | 0.34 | 0.41 | 3.16 |
| 63 | E-ESS | 74.7 | 172.4 | 67 | 7.4 | -24.4 | | 0.08 | 0.43 | 0.36 | 0.47 | 0.39 | 0.98 |
| 66 | E-ESS | 75.9 | 174.3 | 20 | 6.7 | -27.4 | -240 | 0.20 | 0.60 | 0.48 | 0.33 | 0.41 | 1.45 |
| *POC-sub* | | | | | | | | | | | | | |
| 13 | LS | 76.8 | 125.9 | 38 | 6.7 | -29.0 | | | | | | | |
| 14 | LS | 76.9 | 127.8 | 47 | 6.0 | -30.9 | | 0.36 | 0.60 | 0.89 | 0.30 | 0.57 | 2.32 |
| 23 | LS | 76.2 | 129.3 | 99 | 11.7 | -27.2 | | | | | | | |
| 25 | LS | 76.0 | 130.7 | 130 | 10.1 | -26.8 | | 1.14 | 0.79 | 0.22 | 0.33 | 0.49 | 0.78 |
| 27 | LS | 76.9 | 132.2 | 54 | 7.9 | -26.3 | | | | | | | |
| 28 | LS | 77.3 | 134.8 | 48 | 6.6 | -28.1 | -365 | 0.87 | 0.49 | 0.86 | 0.39 | 0.71 | 3.01 |
| 29 | LS | 77.8 | 136.7 | 60 | 6.4 | -25.9 | | | | | | | |
| 39 | W-ESS | 77.7 | 141.4 | 93 | 6.9 | -26.7 | | | | | | | |
| 40 | W-ESS | 77.6 | 145.8 | 52 | 7.2 | -26.4 | | 0.65 | 0.28 | 0.29 | 0.29 | 0.49 | 1.86 |
| 41 | W-ESS | 77.0 | 148.3 | 170 | 11.3 | -27.5 | | | | | | | |
| 44 | W-ESS | 76.3 | 146.0 | 95 | 6.5 | -28.1 | -193 | 0.37 | 0.37 | 0.47 | 0.37 | 0.41 | 2.31 |
| 46 | W-ESS | 76.4 | 149.9 | 170 | 5.8 | -25.8 | | | | | | | |
| 48 | W-ESS | 76.5 | 150.8 | 78 | 6.1 | -26.2 | | 0.31 | 0.36 | 0.47 | 0.25 | 0.42 | 1.60 |
| 50 | W-ESS | 75.8 | 158.5 | 170 | 6.3 | -28.4 | | 0.28 | 0.31 | 0.35 | 0.27 | 0.39 | 1.06 |
| 52 | E-ESS | 74.1 | 160.6 | 72 | 6.4 | -27.1 | -258 | 0.34 | 0.24 | 0.55 | 0.43 | 0.51 | 1.97 |
| 56 | E-ESS | 74.6 | 161.9 | 120 | 8.9 | -25.9 | | | | | | | |
| 59 | E-ESS | 74.4 | 168.5 | 120 | 7.6 | -26.4 | | | | | | | |
| 60 | E-ESS | 73.4 | 169.5 | 360 | 7.4 | -27.5 | -185 | 0.17 | 0.69 | 0.91 | 0.27 | 0.42 | 1.76 |
| 63 | E-ESS | 74.7 | 172.4 | 170 | 7.0 | -25.0 | | 0.17 | 0.59 | 0.63 | 0.23 | 0.34 | 1.88 |

[1]POC concentrations ($\mu$g·L$^{-1}$)
[2]Lignin OC-normalized concentrations (mg·g$^{-1}$ OC)




**Table 3. Composition of surface sediment samples collected in the outer Eurasian Arctic Shelf.**

| ID | Region | Lat | Long | Depth[1] | SOC[2] | OC/TN | $\delta^{13}C$ | $\Delta^{14}C$ | Lignin[3] | S/V | C/V | Sd/Sl | Vd/Vl | 3.5Bd/V |
|----|--------|-----|------|-------|------|-------|------|------|--------|-----|-----|-------|-------|---------|
| *SOC* | | | | | | | | | | | | | | |
| 1 | LS | 78.9 | 125.2 | -3120 | 1.0 | 7.1 | -22.3 | -418 | 0.58 | 0.72 | 0.34 | 0.62 | 0.89 | 0.62 |
| 4 | LS | 77.8 | 126.7 | -2186 | 1.3 | 6.8 | -22.5 | -428 | 0.37 | 0.60 | 0.39 | 0.95 | 1.27 | 1.29 |
| 6 | LS | 77.1 | 127.4 | -92 | 0.8 | 6.7 | -23.2 | | | | | | | |
| 13 | LS | 76.8 | 125.9 | -74 | 1.3 | 7.4 | -24.1 | | | | | | | |
| 14 | LS | 76.9 | 127.8 | -64 | 0.9 | 6.4 | -24.3 | -314 | 0.77 | 0.56 | 0.24 | 0.90 | 1.21 | 0.66 |
| 23 | LS | 76.2 | 129.3 | -56 | 1.6 | 7.6 | -25.0 | -333 | 0.62 | 0.53 | 0.24 | 0.99 | 1.33 | 0.65 |
| 24 | LS | 75.6 | 129.6 | -46 | 1.1 | 6.9 | -24.8 | -284 | 1.82 | 0.56 | 0.30 | 0.95 | 1.26 | 0.51 |
| 25 | LS | 76.0 | 130.7 | -53 | 1.6 | 8.4 | -25.5 | | 2.35 | 0.56 | 0.31 | 0.81 | 0.95 | 0.34 |
| 26 | LS | 76.5 | 132.0 | -52 | 1.2 | 7.9 | -24.4 | -441 | | | | | | |
| 27 | LS | 76.9 | 132.2 | -44 | 1.4 | 7.5 | -24.2 | | | | | | | |
| 28 | LS | 77.3 | 134.8 | -49 | 1.4 | 7.1 | -23.8 | -421 | 1.00 | 0.53 | 0.27 | 1.14 | 1.43 | 0.57 |
| 29 | LS | 77.8 | 136.7 | -57 | 1.1 | 6.9 | -23.4 | -427 | | | | | | |
| 39 | W-ESS | 77.7 | 141.4 | -45 | 0.5 | 7.9 | -24.0 | | 0.77 | 0.48 | 0.20 | 0.91 | 1.17 | 0.59 |
| 40 | W-ESS | 77.6 | 145.8 | -47 | 0.4 | 7.1 | -23.7 | -457 | 0.74 | 0.64 | 0.23 | 0.96 | 1.26 | 0.72 |
| 41 | W-ESS | 77.0 | 148.3 | -40 | 0.3 | 7.7 | | | | | | | | |
| 44 | W-ESS | 76.3 | 146.0 | -43 | 1.2 | 7.9 | -24.8 | -484 | 1.41 | 0.55 | 0.39 | 0.73 | 0.97 | 0.51 |
| 45 | W-ESS | 76.4 | 148.1 | -40 | 1.0 | 7.7 | -24.4 | | | | | | | |
| 46 | W-ESS | 76.4 | 149.9 | -40 | 1.1 | 7.2 | -24.7 | -463 | | | | | | |
| 48 | W-ESS | 76.5 | 150.8 | -40 | 1.4 | 7.4 | -25.8 | -345 | 0.28 | 0.50 | 0.29 | 1.17 | 1.23 | 1.26 |
| 49 | W-ESS | 76.5 | 156.9 | -47 | 1.3 | 6.6 | -23.6 | -375 | 0.17 | 0.56 | 0.32 | 1.31 | 1.96 | 1.89 |
| 50 | W-ESS | 75.8 | 158.5 | -44 | 1.2 | 6.7 | -24.6 | -523 | 0.54 | 0.50 | 0.21 | 0.92 | 1.20 | 0.69 |
| 52 | E-ESS | 74.1 | 160.6 | -46 | 0.8 | 7.1 | -23.9 | -550 | 0.29 | 0.44 | 0.16 | 0.76 | 0.95 | 0.88 |
| 56 | E-ESS | 74.6 | 161.9 | -48 | 1.1 | 7.3 | -23.7 | | | | | | | |
| 57 | E-ESS | 74.4 | 163.7 | -52 | 1.6 | 7.0 | -24.2 | -326 | 0.12 | 0.56 | 0.31 | 0.80 | 1.14 | 1.92 |
| 58 | E-ESS | 74.4 | 166.2 | -54 | 1.7 | 7.5 | -23.8 | -296 | 0.10 | 0.72 | 0.52 | 0.78 | 1.43 | 2.38 |
| 59 | E-ESS | 74.4 | 168.5 | -54 | 1.7 | 6.7 | -23.5 | -307 | | | | | | |
| 60 | E-ESS | 73.4 | 169.5 | -43 | 0.9 | 7.6 | -24.0 | -472 | 0.74 | 1.05 | 0.56 | 0.70 | 0.76 | 0.54 |
| 61 | E-ESS | 74.1 | 170.9 | -51 | 1.8 | 7.0 | -24.2 | -318 | 0.25 | 0.85 | 0.81 | 0.91 | 1.03 | 0.86 |
| 63 | E-ESS | 74.7 | 172.4 | -64 | 1.7 | 7.3 | -22.7 | -251 | 0.10 | 0.81 | 0.83 | 0.66 | 1.14 | 1.87 |
| 66 | E-ESS | 75.9 | 174.3 | -239 | 0.8 | 5.6 | -21.0 | -448 | 0.10 | 0.76 | 0.41 | 0.34 | 0.53 | 1.38 |

[1]Water depth (m)
[2]Percentage of sedimentary organic carbon
[3]Lignin OC-normalized concentrations (mg·g⁻¹ OC)













**Figures**

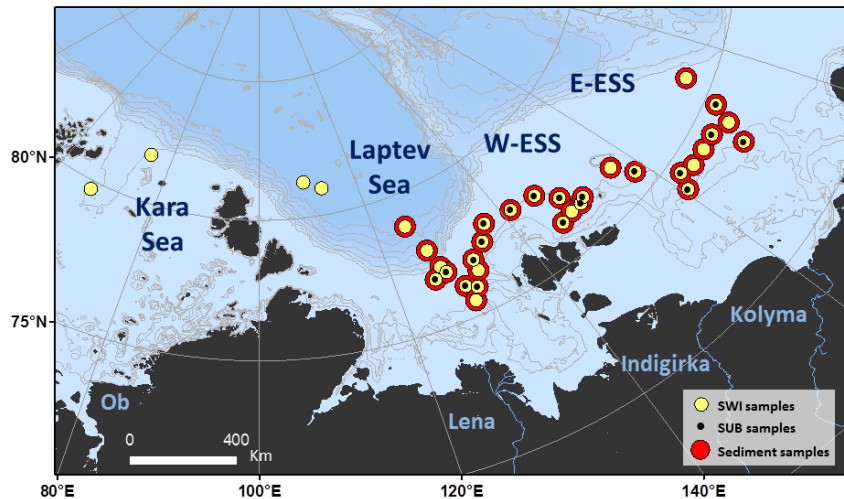


**Figure 1.** Map of the Eurasian Arctic Shelf including the Kara, Laptev and East Siberian seas
(E-ESS, eastern East Siberian Sea; W-ESS, western East Siberian Sea). SWI samples,
seawater intake samples (surface water samples at 8 m depth); SUB samples, samples
obtained by submersible pump (near-bottom water samples, 5 m above bottom).





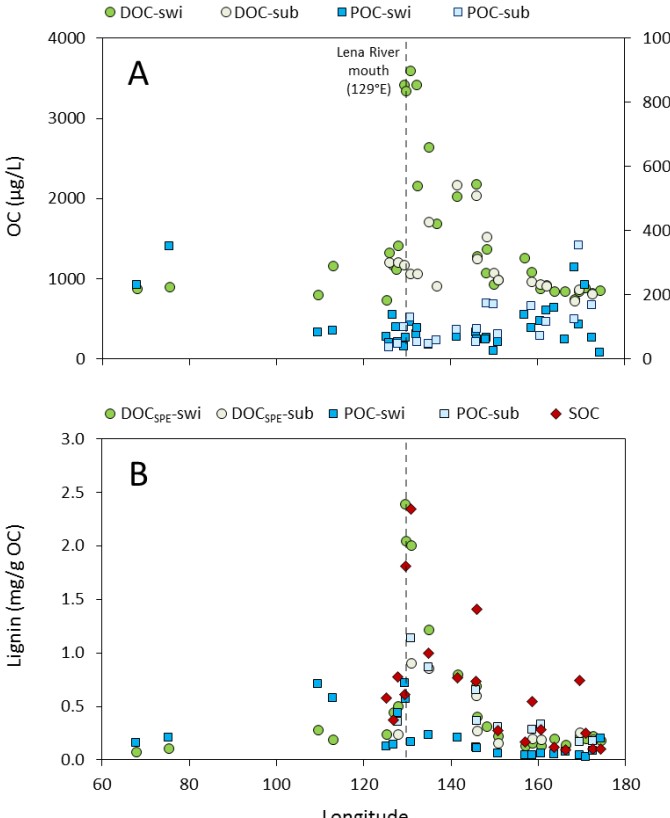


**Figure 2.** Longitudinal distribution of organic matter content in water and sediment samples. A) Organic carbon concentrations (μg/L) in DOC (green circles) and POC (blue squares). B) Lignin concentrations (mg/g OC) in DOC$_{SPE}$ (green circles), POC (blue squares) and SOC (red diamonds); swi, seawater intake samples (surface water samples at 8 m depth); sub, samples obtained by submersible pump (near-bottom water samples, 5 m above bottom). Dash line indicate the latitude of the Lena River mouth.







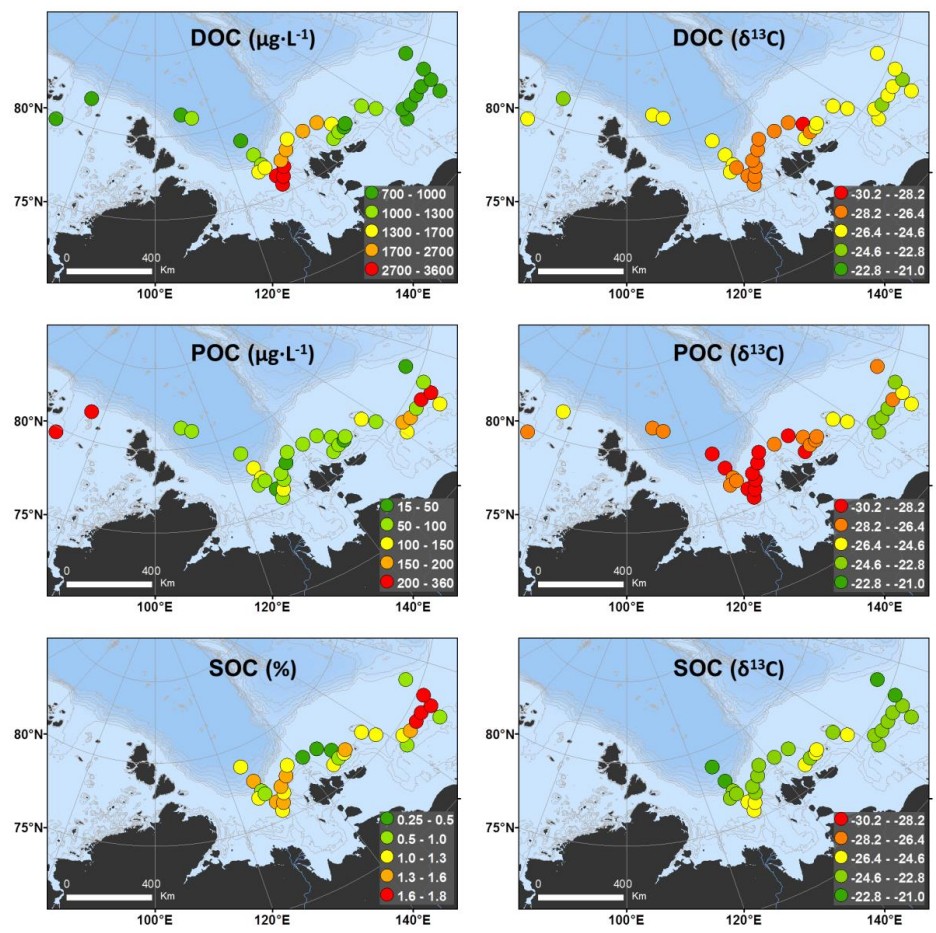



**Figure 3.** Spatial distribution of organic carbon concentrations and $\delta^{13}$C signatures in the
DOC, POC (surface water samples at 8 m depth) and SOC pools of the ESAS.











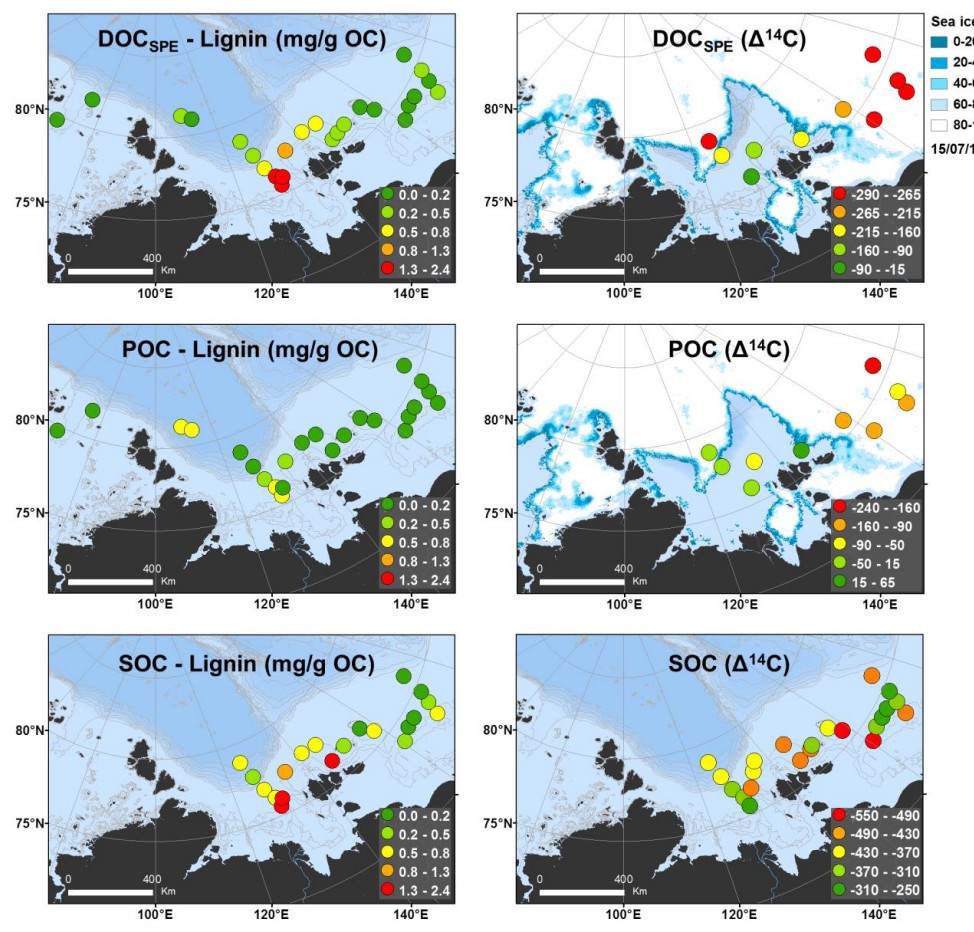



**Figure 4.** Spatial distribution of lignin (mg/g OC) and $\Delta^{14}$C signatures in the DOC$_{SPE}$, POC
(surface water samples at 8 m depth) and SOC pools of the ESAS. Sea ice (%) during the first
sampling day in the ESAS (15/7/2015).











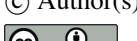

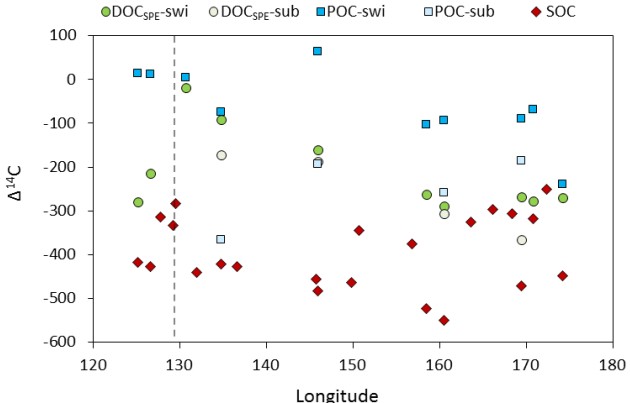



**Figure 5.** Longitudinal distribution of $\Delta^{14}C$ signatures in $DOC_{SPE}$ (green circles), POC (blue squares) and SOC (red diamonds); swi, sea water intake samples (surface water samples at 8 m depth); sub, samples obtained by submersible pump (near-bottom water samples, 5 m above bottom). Dash line indicate the latitude of the Lena River mouth.






















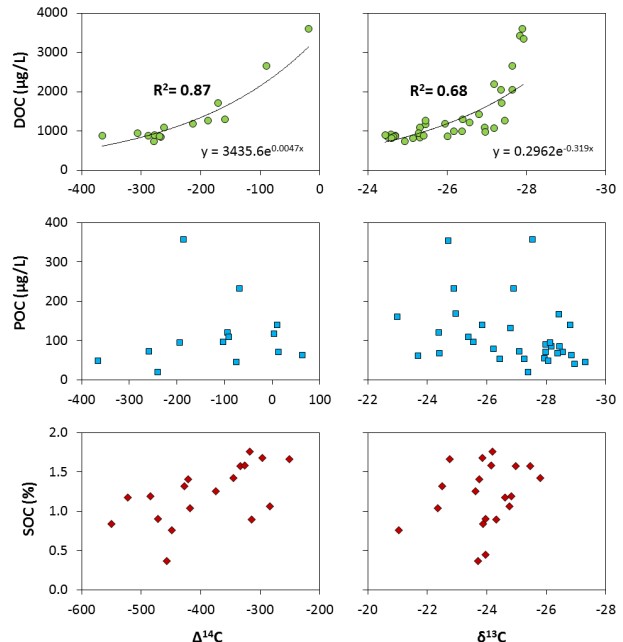

**Figure 6.** Relationships between organic carbon and $\Delta^{14}$C and $\delta^{13}$C signatures in the DOC (green circles), POC (blue squares) and SOC (red diamonds).



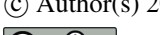

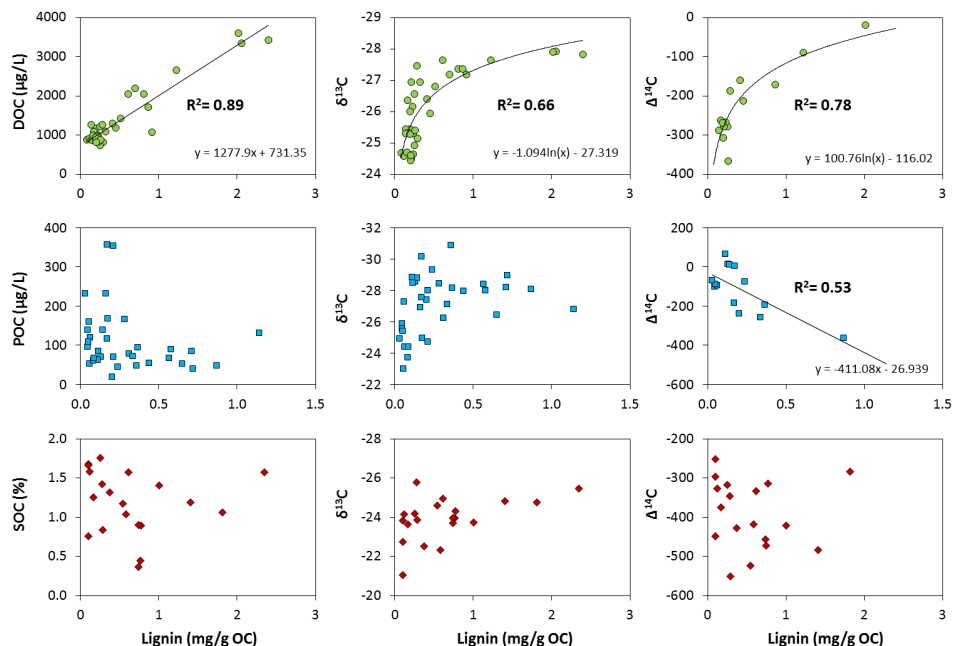



**Figure 7.** Correlations between lignin concentrations (mg/g OC) and organic carbon, $\delta^{13}$C
and $\Delta^{14}$C signatures in DOC (green circles), POC (blue squares) and SOC (red diamonds).





















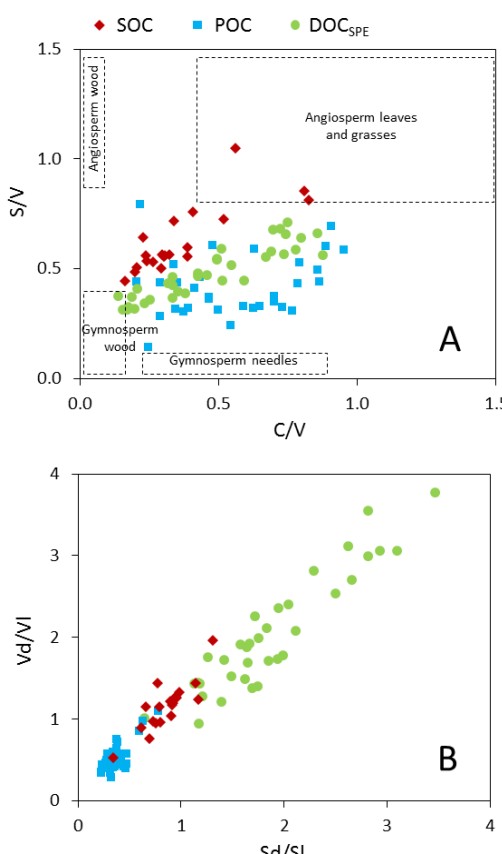


**Figure 8.** Lignin-phenols ratios in DOC$_{SPE}$ (green circles), POC (blue squares) and SOC (red
diamonds). A) Classical source plot of syringyl/vanillyl (S/V) vs. cinnamyl/vannillyl (C/V).
B) The acid/aldehyde ratios of syringyl (Sd/Sl) vs. vanillyl (Vd/Vl).















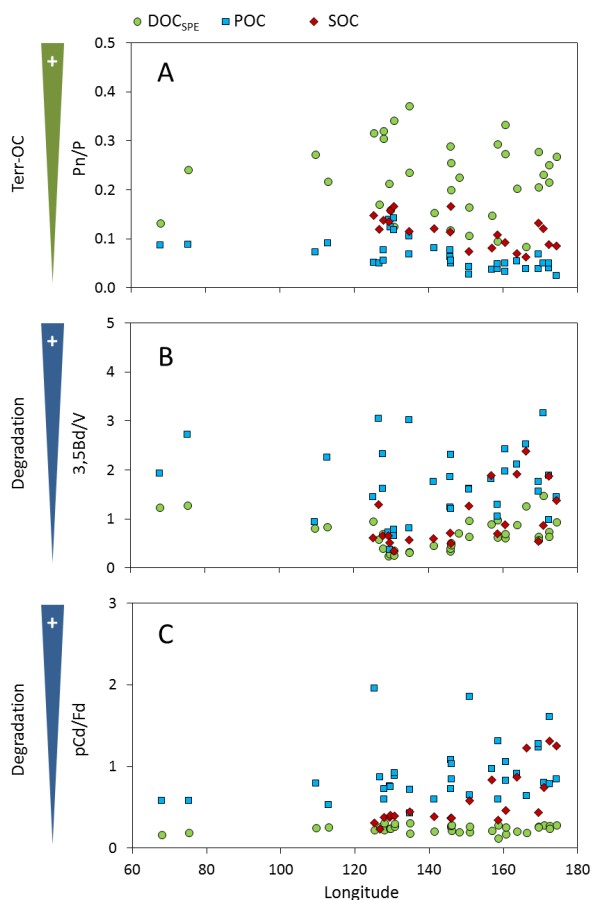



**Figure 9.** Lignin proxies of Terr-OC and relative degradation state of DOC$_{SPE}$ (green circles), POC (blue squares) and SOC (red diamonds). A) hydroxyacetophenone/$p$-hydroxybenzoic acids (Pn/P). B) Ratios between 3,5-dihydroxybenzoic acid and vanillyl phenols (3,5-Bd/V). C) Ratios between $p$-coumaric acid and ferulic acid (pCd/Fd).

963

964

965