# Peer review of "Contrasting composition of terrigenous organic matter in the dissolved, particulate and sedimentary organic carbon pools on the outer East Siberian Arctic Shelf"

_Biogeosciences, 2016_

## Referee Comment (RC1) · Anonymous Referee #1 · 2 Aug 2016

Review of Biogeosciences Discussion bg-2016-260 "Contrasting composition of terrigenous organic matter in the dissolved, particulate and sedimentary organic carbon pools on the outer East Siberian Arctic Shelf" by Salvado et al.,

The manuscript of Salvado et al., provides new data to examine organic carbon cycling in the East Siberian Arctic Shelf. This is an important issue. The Arctic Ocean is undergoing significant warming, and this is projected to increase over the coming century. The impact of this warming on carbon pools remains an important question – i.e. are the carbon pools stable, or could they degrade and release gaseous carbon to

contribute to atmospheric CO2 and CH4 budgets. One of the key unknowns remains the source, age and nature of organic carbon in the water column and at the seafloor (and how these are linked). Despite a large amount of recent work in the study area (involving some of the authors), we are still sample limited, and this paper delivers an impressive new dataset.

A strength of the study is that they examine dissolved and particulate phases in the surface water, and deep water, alongside the sedimentary organic matter. The paper presents bulk elemental and isotopic measurements (including radiocarbon) alongside a range of biomarker yields to help reveal patterns in the source and processing of OC. These aspects allow the authors to examine the contrasting source and age of the OC, and discussion what processes are responsible for these contrasts. Age contrasts between DOC and POC have been examined before, but not too my knowledge through the water column, and in relation to the lignin biomarkers. The decoupling of deep POC and DOC is very stark. The findings are new and should certainly interest the readership at Biogeosciences.

However, I have one main comment, which is comprised of a few parallel issues/observations which I feel the authors should work to clarify in revision:

1. How sensitive are the biomarker proxies (used to examine marine vs terrestrial, and evidence for degradation): a) The Pn/P (hydroxyacetophenone/p-hydroxybenzoic acids) ratio is discussed in the context of a marine versus terrestrial biomarker (Fig. 9a) and used to conclude (stated in the abstract and throughout) that POC is 'mainly composed' of marine OC, suggesting more than 50% of the TOC is marine in origin. However, I think the paper needs to be a little more cautious on this point. They discuss some of the caveats of this biomarker ratio in the text, but also remember this is an extracted component of the TOC. What are the bulk proxies (elemental, isotope) telling us?

Indeed, what is the lignin yield telling us? After two reads, to me, the lignin yield data

doesn't seem to line up with this conclusion. Some surface POC has more lignin yield than the DOC (and not significantly less) and the deep POC samples also have high Lignin yields. Have I missed something here? I don't follow how the lignin yield (a 'major' biomarker if you like) can show these patterns, but the bulk Pn/P ratio tell us something else about organic matter source? The POC is 13C-depleted (note – seems to correlate with DOC d13C looking at the maps) which could be marine organic matter soured from DIC with a remineralisation signature (as the authors explain somewhere). Or it could be the 'top soil' end member identified by Vonk et al., 2012, Nature, at -26 to -30 per mil (see the supplement). Added to this, the 14C-depleted nature of the bulk, deep water POC is easier to explain if a 'permafrost' terrestrial OC signature, so again, how can this be material dominated by marine OC?

b) I have a related comment on the degradation proxies. I think these are important to pursue, so they should certainly be in here and discussed. But, there are some key questions. Do we know the starting compositions of organic matter? The authors acknowledge we have no POC samples from terrestrial materials to compare too. Second, if the biomarkers suggest 'significant' degradation (or as the authors put it, more than we have seen before), should we not also see this in the bulk %OC too? Is there a link in this dataset, or is the %OC too variable due to other factors (sorting, heterogeneity). Indeed, for the SOC samples, a quick look in Table 3 reveals those with high 3,5Bd/V (1.87, 1.89, 2.38) appear to have relatively high %OC (1.7%, 1.3%, 1,7% respectively), while those with lower %OC (0.4%, 0.5%) have lower 3,5Bd/V (0.72, 0.59). This is the opposite of what you might expect for degradation, i.e. higher 3,5Bd/V should indicate more degradation and lower %OC. Does this suggest a strong role of a source change, rather than a degradation signal, to explain the 3,5Bd/V ratios?

More discussion on these issues, and awareness of the caveats would benefit the manuscript. I wonder if a combined results/discussion is helping this. It could be wise to separate these sections to clarify some of these points.

Other comments (by line numbers)

37: based on my reading of the manuscript and the data, a key contrast is that the lignin is highest in the oldest POC samples, whereas the lignin is highest in the youngest DOC samples, suggesting a significant decoupling of terrestrial OC sources and pathways. I was less convinced by the evidence for 'preferential' (implies >50% ?) marine origin of the POC (especially at the seafloor).

44: define ESS

36, 47 & 58: these are important observations, but they are repeated here. Consider restructuring the abstract to make the results and implications/discussion more distinct.

45: the sea ice observation (Fig. 4) seems important, but it is not well explained. Perhaps reorganise the abstract and add a sentence to more clearly make this point.

57: how much evidence for this is presented? Perhaps add a caveat about lignin yields not being that different, and %OC being not that different (more bulk indicators of net degradation?)?

60: ended on a rather generic note. The final sentences could do a better job of summing up the main findings.

87: there are two issues here. 1st the potential fate of OC at the seafloor due to warming (we know nothing about this?!) and 2nd the changing terrestrial inputs. The text could be modified to make these points.

100: define acronyms at first use here.

109: Line 102 mentions the Arctic Ocean here, but I think these comments are linked to the Eurasian Shelf. Certainly, the POC does not appear to degrade faster than POC in the beaufort Sea/Mackenzie Delta studies, probably because of the much higher sediment input at this point. Anyhow, worth clarifying.

116: another issue worth flagging is that the POC in the Eurasian rivers is not well characterised. Previous studies (Vonk et al and others) rely on estuarine samples.

127: could be useful to provide estimates of these OC fluxes.

137: I agree, this is cool, but could you better explain why it is important to do this.

153: how much? (% or Tg C yr-1)?

201: to what depth was the sediment sampled? I wasn't too clear on whether these were surface samples, or cores

221: HCl fumigation (vapour) or leach (liquid) method?

304: this DOC D14C is much older than reported in the study of Eurasian Arctic Rivers by Raymond et al., 2007, GBC, who have values D14C > +39 per mil in the Lena! Worth discussion.

322-327: This seems like an important observation and worth expanding on (certainly worth its own paragraph). To me, this is the key evidence for decoupling of the POC and DOC pools, and as the authors point out, the evidence for pre-aged OC in the system.

328-339: this discussion needs to be more clear on whether you are invoking mixing to explain the patterns, or processes, or both.

355: sentence could be clearer.

364-366: interesting. How does this work hydrodynamically? Lignin rich materials may contain more coarser woody particles, which should be buoyant/neutrally buoyant. This suggests not, and indicates waterlogged terr-OC which can sink. This is seen in the Mackenzie River (but we don't have samples from the Eurasian rivers to examine).

374: 'addition' rather than 'dilution'?

394: split the paragraph here. These are important observations which need to be better drawn out.

437-438: this sentence doesn't fit well with the discussion on 480, where V is noted to

be sensitive to degradation. So, could the shift in S/V space for the POC be due to this, and not changing source? Why is the sediment not V depleted?

454: I don't understand how this can be the case ('mainly composed' implies that >50% OC is marine) when the lignin yields (Fig 2) are higher in some POC samples than for DOC. Doesn't that suggest the same lignin loading, and similar contributions from terrestrial OC in POC and DOC?

455 & 526: role of sea ice and pacific inflow – when I read these bits of text, I thought it was not well supported by the data. After a second reading and closer look at Figure 4, I see your point. This needed to be better explained (why would sea ice do this?) and perhaps drawn out to give this its own paragraph in the discussion.

455: missing text

472: are these Arctic studies or global – clarify.

477- how sensitive are these ratios? are they a linear function of degradation? The second point is important when interpreting what a change from 1-3 actually means.

519-: Overall I found the conclusions didn't map as well onto the discussion and data as they could have done.

527: seemed to be a somewhat misleading statement based on the available data. It would be better to explain the key findings first, and discussion what they indicate, with caveats.

531: the S/V ratios show a mixture in fig. 8a? not a dominant source from one end member??

541: what do you mean by 'less' burial and 'more' mineralisation. Vonk et al., (and others) already indicate this is going on. So, do you mean the numbers could be wrong by a lot (i.e. 50%) or a little (1-5%), or we don't know.

Fig 2b – my comment about lignin yields is contained here (1a above). Subwater and

surface POC can have higher lignin yields than DOC, but the main text then concludes a mainly marine composition for POC – I don't follow that logic.

Fig 5- nice data. Could you plot (a) and (b), with DOC one lot, and POC on the other, just to help see the surface-depth contrast in the POC.

Fig 6 – why exponential functions?

Fig. 8. Plot the surface/subsurface distinction here.

Fig. 9. Do we know how sensitive are these proxies are? What are their expected compositions for marine and terrestrial OC, and how much do they vary. In other words, when we have +ve, are we talking about change of a few % of the total OC, or a lot more? And/or can the source of the OC impact the ratios. This is discussed in the main text, but the abstract and conclusions make quite assertive points based on these data. See comment 1b above.

---

## Author Comment (AC1) · 18 Aug 2016

**Author responses to referee 1 comments, and planned edits to BG ms "*Contrasting composition of terrigenous organic matter in the dissolved, particulate and sedimentary organic carbon pools on the outer East Siberian Arctic Shelf*" (doi:10.5194/bg-2016-260, 2016)**

by Salvadó, Tesi, Sundbom, Karlsson, Kruså, Semiletov, Panova and Gustafsson

We thank the reviewer for her/his careful reading of our manuscript. The constructive reviews and suggestions contribute to improve the paper further during our revisions. All referee comments and our responses, as well as the planned edits, are detailed below, organized such that first the reviewer comments are given in italic, directly followed by our response and outline of planned edits in regular font.

*Reviewer Comments*

*The manuscript of Salvado et al., provides new data to examine organic carbon cycling in the East Siberian Arctic Shelf. This is an important issue. The Arctic Ocean is undergoing significant warming, and this is projected to increase over the coming century. The impact of this warming on carbon pools remains an important question – i.e. are the carbon pools stable, or could they degrade and release gaseous carbon to contribute to atmospheric $CO_2$ and $CH_4$ budgets. One of the key unknowns remains the source, age and nature of organic carbon in the water column and at the seafloor (and how these are linked). Despite a large amount of recent work in the study area (involving some of the authors), we are still sample limited, and this paper delivers an impressive new dataset.*

*A strength of the study is that they examine dissolved and particulate phases in the surface water, and deep water, alongside the sedimentary organic matter. The paper presents bulk elemental and isotopic measurements (including radiocarbon) alongside a range of biomarker yields to help reveal patterns in the source and processing of OC. These aspects allow the authors to examine the contrasting source and age of the OC, and discussion what processes are responsible for these contrasts. Age contrasts between DOC and POC have been examined before, but not too my knowledge through the water column, and in relation to the lignin biomarkers. The decoupling of deep POC and DOC is very stark. The findings are new and should certainly interest the readership at Biogeosciences.*

We appreciate this positive overall assessment of the value and suitability of our paper.

*However, I have one main comment, which is comprised of a few parallel issues/observations which I feel the authors should work to clarify in revision:*

*1. How sensitive are the biomarker proxies (used to examine marine vs terrestrial, and evidence for degradation): a) The Pn/P (hydroxyacetophenone/p-hydroxybenzoic acids) ratio is discussed in the context of a marine versus terrestrial biomarker (Fig. 9a) and used to conclude (stated in the abstract and throughout) that POC is 'mainly composed' of marine OC, suggesting more than 50% of the TOC is marine in origin. However, I think the paper needs to be a little more cautious on this point. They discuss some of the caveats of this biomarker ratio in the text, but also remember this is an extracted component of the TOC.*

We understand the review point and have now clarified these points. We do not mean to base conclusions only on this Pn/P ratio. We have thus removed Figure 9A to prevent misunderstandings. Throughout the manuscript we have now edited to state that POC in surface waters may have a preferential marine origin, mainly inferred from its very low lignin concentrations, compared to DOC and SOC pools (Figures 2B and 4). Further, we have also corrected that the Pn/P ratios observed in DOC (0.08-0.37), SOC (0.06-0.17) and POC (0.02-0.14) suggest that the OC in the dissolved pool is more terrestrial than in the particulate and sediment pools (Tables 1, 2 and 3; Figure 9A).

Nevertheless, these proxies should be interpreted carefully as P products account for less than 0.1% of the bulk OC so care should be placed when extrapolating the biomarker data..

*What are the bulk proxies (elemental, isotope) telling us? Indeed, what is the lignin yield telling us? After two reads, to me, the lignin yield data doesn't seem to line up with this conclusion. Some surface POC has more lignin yield than the DOC (and not significantly less) and the deep POC samples also have high Lignin yields. Have I missed something here? I don't follow how the lignin yield (a 'major' biomarker if you like) can show these patterns, but the bulk Pn/P ratio tell us something else about organic matter source? The POC is 13C-depleted (note – seems to correlate with DOC d13C looking at the maps) which could be marine organic matter soured from DIC with a remineralisation signature (as the authors explain somewhere). Or it could be the 'top soil' end member identified by Vonk et al., 2012, Nature, at -26 to -30 per mil (see the supplement). Added to this, the 14C-depleted nature of the bulk, deep water POC is easier to explain if a 'permafrost' terrestrial OC signature, so again, how can this be material dominated by marine OC?*

We agree and have now clarified these matters. The bulk proxies and the lignin yield are telling us that there is a significant decoupling of terrestrial OC sources and pathways. While higher lignin values in "young" DOC are related to the fluvial discharge of fresh OC from the Lena River, enhanced lignin yields in "old" POC, from near-bottom waters, most likely reflect the off-shelf transport of permafrost-derived OC in the nepheloid layer (Figures 2B, 3 and 4).

We have now included in the manuscript that some surface POC samples, the ones in the Kara Sea, had more lignin than in the DOC, but those concentrations were relatively low compared to the higher lignin yields observed in the dissolved and sedimentary OC pools from the ESAS. We also explain in many parts of the manuscript that the enhanced lignin values in near-bottom waters may be due to the off-shelf transport of permafrost-derived OC in the nepheloid layer, through repeated cycles of deposition and resuspension across the shelf.

We agree that the Pn/P ratio is not following the same trend as the lignin yield in the particulate and sedimentary OC pools. We have thus clarified that this proxy should be interpreted carefully and removed Figure 9A to prevent misunderstandings. On the other hand, we are confident stating that POC in surface waters of the outer-shelf of the ESAS is composed preferentially of marine OC mainly due to its very low lignin concentrations, compared to DOC and SOC pools (Figures 2 and 4), and its no-correlations with $\delta^{13}$C and lignin values.

*b) I have a related comment on the degradation proxies. I think these are important to pursue, so they should certainly be in here and discussed. But, there are some key questions. Do we know the starting compositions of organic matter? The authors acknowledge we have no POC samples from terrestrial materials to compare too. Second, if the biomarkers suggest 'significant' degradation (or as the authors put it, more than we have seen before), should we not also see this in the bulk %OC too? Is there a link in this dataset, or is the %OC too variable due to other factors (sorting, heterogeneity). Indeed, for the SOC samples, a quick look in Table 3 reveals those with high 3,5Bd/V (1.87, 1.89, 2.38) appear to have relatively high %OC (1.7%, 1.3%, 1,7% respectively), while those with lower %OC (0.4%, 0.5%) have lower 3,5Bd/V (0.72, 0.59). This is the opposite of what you might expect for degradation, i.e. higher 3,5Bd/V should indicate more degradation and lower %OC. Does this suggest a strong role of a source change, rather than a degradation signal, to explain the 3,5Bd/V ratios?*

We understand the review point; however, we cannot assume that higher 3,5Bd/V necessarily correlates with lower %OC. For example, OC can be affected by several other factors which include grain-size and dilution with marine OC. Therefore, the OC content itself does not necessarily reflect the extent of degradation as the 3,5Bd/V ratio does.

Nevertheless, we agree and have now included in the text that 3,5Bd/V could support the role of a source change as observed with S/V and C/V proxies. In this data set pCd/Fd ratios, another ratio that has been used as a diagenetic indicator (Amon et al., 2012; Houel et al., 2006), followed the same

pattern as the ones observed in 3,5-Bd/V ratios with higher values in POC and SOC (Tables 1, 2 and 3; Figure 9C). This would strengthen the hypothesis that POC and SOC are more degraded than DOC.

*More discussion on these issues, and awareness of the caveats would benefit the manuscript. I wonder if a combined results/discussion is helping this. It could be wise to separate these sections to clarify some of these points.*

We have now strengthened the discussion of these aspects in the manuscript.

*Other comments (by line numbers)*

*37: based on my reading of the manuscript and the data, a key contrast is that the lignin is highest in the oldest POC samples, whereas the lignin is highest in the youngest DOC samples, suggesting a significant decoupling of terrestrial OC sources and pathways. I was less convinced by the evidence for 'preferential' (implies >50% ?) marine origin of the POC (especially at the seafloor).*

We agree with this assessment and have now deleted this sentence here in the abstract. Moreover, we have now clarified throughout the manuscript that POC in surface waters may have a preferential marine origin mainly due to its very low lignin values. We also state now in many parts of the manuscript that the higher lignin values in younger DOC samples are likely related to the fluvial discharge of fresh OC from the Lena River, while the higher lignin yields in older POC samples, from near-bottom waters, is bound to reflect the off-shelf transport of permafrost-derived OC in the nepheloid layer.

*44: define ESS*

We have now deleted this acronym as we have restructured the abstract.

*36, 47 & 58: these are important observations, but they are repeated here. Consider restructuring the abstract to make the results and implications/discussion more distinct.*

We agree and have now restructured the abstract. We have now deleted the sentences "Depleted $\delta^{13}$C, modern $\Delta^{14}$C and lignin phenols concentrations were all well correlated with DOC levels indicating a relatively young terrestrial contribution. In contrast, POC may have a preferential marine origin, as its concentrations were not correlated with isotope and terrestrial biomarker proxies" and "Overall, DOC is strongly affected by the Lena River plume transporting young Terr-OC from topsoil and/or recently produced vascular pant material, while near-bottom POC and SOC preferentially carries off-shelf old OC released from thawing permafrost". Further, we have now included the sentence "Overall, the key contrast between enhanced lignin yields both in the youngest DOC and the oldest POC samples suggest a significant decoupling of terrestrial OC sources and pathways."

*45: the sea ice observation (Fig. 4) seems important, but it is not well explained. Perhaps reorganise the abstract and add a sentence to more clearly make this point.*

We have now restructured the abstract and rewritten this sentence "Further, the sea ice coverage and the Pacific inflow from the east seem to have a strong influence on those carbon pools, which present older and more enriched δ13C signatures under the sea ice extent."

*57: how much evidence for this is presented? Perhaps add a caveat about lignin yields not being that different, and %OC being not that different (more bulk indicators of net degradation?)?*

We agree that there in not a strong evidence for this; therefore, we have now deleted this sentence "This suggests that the remobilized old OC from thawing permafrost, which is mainly transported within these pools, could experience less burial and more mineralization than believed earlier." Moreover, we have included the sentence "suggesting more degradation within these pools".

*60: ended on a rather generic note. The final sentences could do a better job of summing up the main findings.*

We have now rewritten this sentence "Overall, the key contrast between enhanced lignin yields both in the youngest DOC and the oldest POC samples reflects a significant decoupling of terrestrial OC sources and pathways."

*87: there are two issues here. 1st the potential fate of OC at the seafloor due to warming (we know nothing about this?!) and 2nd the changing terrestrial inputs. The text could be modified to make these points.*

We have now included these points: "The sources and the inputs of Terr-OC are likely to vary in the northern shelf margin due to the changing climate".

*100: define acronyms at first use here.*

We have now defined DOC, POC and SOC here.

*109: Line 102 mentions the Arctic Ocean here, but I think these comments are linked to the Eurasian Shelf. Certainly, the POC does not appear to degrade faster than POC in the beaufort Sea/Mackenzie Delta studies, probably because of the much higher sediment input at this point. Anyhow, worth clarifying.*

This appears to be a misunderstanding. Here we are referring to studies in different places of the Arctic Ocean that indicated a conservative behavior of DOC.

*116: another issue worth flagging is that the POC in the Eurasian rivers is not well characterised. Previous studies (Vonk et al and others) rely on estuarine samples.*

We have now clarified this and state that POC in the Eurasian rivers is not well characterised.

*127: could be useful to provide estimates of these OC fluxes.*

We have now included an estimate of Terr-OC input in the ESAS "(22±8 Tg OC/yr; Vonk et al., 2012)".

*137: I agree, this is cool, but could you better explain why it is important to do this.*

We have now explained why it is important to do this study "…seeking to test the hypothesis that carbon pools may carry different types of OC with different propensity toward off-shelf transport and degradation".

*153: how much? (% or Tg C yr-1)?*

We have now included the percentage of ICD-PF that enters the Laptev Sea (53±5%; Vonk et al., 2012).

*201: to what depth was the sediment sampled? I wasn't too clear on whether these were surface samples, or cores*

We have now clarified that "Sediment cores were collected from 40 to 3120 m water depth" and that "This study focused on surface sediments (0-1 cm)."

*221: HCl fumigation (vapour) or leach (liquid) method?*

We have now clarified that "surface sediment samples were subsampled and subject to aqueous acidification with HCl (1.5M)"

*304: this DOC D14C is much older than reported in the study of Eurasian Arctic Rivers by Raymond et al., 2007, GBC, who have values D14C > +39 per mil in the Lena! Worth discussion.*

We agree and have now included "The radiocarbon signatures in the DOC pool of the outer-shelf of the ESAS are much older than those observed in the Lena River (>39‰) (Raymond et al., 2007), but similar and even younger than those reported in surface waters of the Canada Basin (<-216‰) (Arctic Ocean), likely reflecting the inputs of Pacific waters."

*322-327: This seems like an important observation and worth expanding on (certainly worth its own paragraph). To me, this is the key evidence for decoupling of the POC and DOC pools, and as the authors point out, the evidence for pre-aged OC in the system.*

We have now expanded these statements "It is important to point out that near-bottom waters presented more depleted and similar $\Delta^{14}$C signatures in both $DOC_{SPE}$ and POC (-258±94‰ and -250±83‰, respectively) than in surface waters (-213±93‰ and -57±86‰, respectively) (Figure 5; Tables 1 and 2), suggesting the same older and terrigenous source of OC in both pools. Those contrasting age offsets between surface and near-bottom waters, particularly for the POC fraction, may reflect the off-shelf transport of OC translocated over long distances from thawing permafrost."

*328-339: this discussion needs to be more clear on whether you are invoking mixing to explain the patterns, or processes, or both.*

We have now clarified this discussion stating that both mixing and processes of the terrigenous DOC along the off-shelf transport explain the correlation between $\delta^{13}$C and DOC ($r^2$=0.68). Further, the relationship between $\Delta^{14}$C and DOC ($r^2$=0.87) represents both the source of the terrigenous DOC and mixing between components.

*355: sentence could be clearer.*

We have now clarified this sentence "Lignin concentrations exhibited contrasting offsets between surface and near-bottom waters, particularly in the POC pool."

*364-366: interesting. How does this work hydrodynamically? Lignin rich materials may contain more coarser woody particles, which should be buoyant/neutrally buoyant. This suggests not, and indicates waterlogged terr-OC which can sink. This is seen in the Mackenzie River (but we don't have samples from the Eurasian rivers to examine).*

We appreciate this assessment; however, vascular plant debris does not float when vascular plant gets old. Actually, it has a high settling velocity because of its dimension (i.e. coarse) (see Tesi et al., 2016). Wood floats because of its air inside that fills all the conducts (e.g., xylem and floem vessels). An old debris floats until its vessels are replaced with water, then it sinks and behaves hydraulically as sandy material. The lignin we see in the outer-shelf is adsorbed on the fine fraction traveling across the margin (see Tesi et al, 2016).

*374: 'addition' rather than 'dilution'?*

We think it is better to state "concentrations in DOC, POC and SOC pools may be interpreted to result from dilution with marine organic during transport".

*394: split the paragraph here. These are important observations which need to be better drawn out.*

We have now split the paragraph into two.

*437-438: this sentence doesn't fit well with the discussion on 480, where V is noted to be sensitive to degradation. So, could the shift in S/V space for the POC be due to this, and not changing source? Why is the sediment not V depleted?*

We understand the review point; however, the ratio cinnamyl over vanillyl (C/V) is just used to distinguish woody lignin from other sources (Goni and Hedges, 1992; Hedges and Mann, 1979). Degradation lowers both S/V and C/V ratio but largely C/V. If degradation was indeed the main process I would expect a sort of linear correlation between S/V and C/V. Therefore, while some degradation cannot be denied, differences in lignin phenols between carbon pools likely reflect a different source.

*454: I don't understand how this can be the case ('mainly composed' implies that >50% OC is marine) when the lignin yields (Fig 2) are higher in some POC samples than for DOC. Doesn't that suggest the same lignin loading, and similar contributions from terrestrial OC in POC and DOC?*

We agree and have now rewritten these statements: "These ratios are in agreement with the relationships of DOC and $\Delta^{14}C$ and $\delta^{13}C$ presented above, which indicate that the DOC exported off-shelf is mainly "young" and terrestrial. Nevertheless, it is important to note that these proxies should be interpreted carefully as P products account for less than 0.1-0.2% of the bulk OC, and in some samples the particulate and sedimentary pools have more lignin yield than the dissolved OC".

*455 & 526: role of sea ice and pacific inflow – when I read these bits of text, I thought it was not well supported by the data. After a second reading and closer look at Figure 4, I see your point. This needed to be better explained (why would sea ice do this?) and perhaps drawn out to give this its own paragraph in the discussion.*

We have now better explained the role of the sea ice and the pacific inflow: "The older and enriched $\delta^{13}C$ signatures in the outer-shelf of the ESS may reflect the influence of sea ice coverage and the Pacific inflow from the East. We suggest that the sea ice would work as a barrier preventing the direct terrigenous input from land and reinforcing the influence of Pacific waters". The influence of the Pacific inflow from the East has also been observed in several other studies (i.e. Semiletov et al., 2005; Stein and Macdonald, 2004; Karlsson et al., 2015).

*455: missing text*

We have now removed this text "Overall, these proxies"

*472: are these Arctic studies or global – clarify.*

We have now clarified that these are global studies.

*477- how sensitive are these ratios? are they a linear function of degradation? The second point is important when interpreting what a change from 1-3 actually means.*

We use these ratios just to estimate which carbon pool is more degraded. They are not a linear function of degradation. Further, these ratios should be interpreted carefully as these products account for less than 0.1% of the bulk OC.

*519-: Overall I found the conclusions didn't map as well onto the discussion and data as they could have done.*

We have now rewritten some parts of the conclusions section:

"Near-bottom waters present more depleted $\Delta^{14}C$ signatures and higher concentrations of lignin, particularly within the POC pool. This reflects the off-shelf transport of thawed-out permafrost OC in the nepheloid layer, through repeated cycles of deposition and resuspension across the shelf."

"Moreover, the opposite trends in the Pn/P and P/V ratios also indicate the terrigenous source of DOC".

"If this hypothesis is true, the remobilized OC from permafrost, which is mainly transported within these carbon pools, could experience less burial and more mineralization than the DOC pool."

"The high abundance of Terr-OC in the outer ESAS, particularly in the dissolved and sedimentary carbon pools, is a clear indicator of the magnitude of shelf to basin transport. Overall, the results are a key evidence for decoupling of the DOC, POC and SOC pools in the ESAS and elucidate the off-shelf transport of permafrost-derived OC in the particulate pool of near-bottom waters."

*527: seemed to be a somewhat misleading statement based on the available data. It would be better to explain the key findings first, and discussion what they indicate, with caveats.*

We agree and have now removed this statement. We have also rewritten the following sentences to "Near-bottom waters present more depleted $\Delta^{14}C$ signatures and higher concentrations of lignin, particularly within the POC pool. This is a key evidence for decoupling of the POC and DOC pools and reflects the off-shelf transport of permafrost-derived OC in the nepheloid layer, through repeated cycles of deposition and resuspension across the shelf."

*531: the S/V ratios show a mixture in fig. 8a? not a dominant source from one end member??*

The ratios of S/V indicate gymnosperm vegetation as the most important source of lignin (Figure 8A), and increasing S/V ratios in the easternmost samples reflect a relatively higher source contribution of tundra plants.

*541: what do you mean by 'less' burial and 'more' mineralisation. Vonk et al., (and others) already indicate this is going on. So, do you mean the numbers could be wrong by a lot (i.e. 50%) or a little (1-5%), or we don't know.*

We have now clarified that regarding 3,5-Bd/V and pCd/Fd ratios, which are higher in the SOC and POC pools, the remobilized OC from permafrost (mainly transported within these carbon pools) could experience less burial and more mineralization than the DOC pool.

*Fig 2b – my comment about lignin yields is contained here (1a above). Subwater and surface POC can have higher lignin yields than DOC, but the main text then concludes a mainly marine composition for POC – I don't follow that logic.*

We have now clarified throughout the manuscript that some surface POC samples, the ones in the Kara Sea, had a larger lignin fraction than that in the DOC, but those Kara Sea lignin concentrations were relatively low compared to the higher lignin yields observed in the dissolved and sedimentary OC pools from the ESAS.

*Fig 5- nice data. Could you plot (a) and (b), with DOC one lot, and POC on the other, just to help see the surface-depth contrast in the POC.*
We have now plotted $\Delta^{14}C$ of POC and SOC in Figure 5A and DOC and SOC in Figure 5B.

*Fig 6 – why exponential functions?*
We observed that our points followed exponential functions.

*Fig. 8. Plot the surface/subsurface distinction here.*

We prefer not to modify these figures. We only want to differentiate the DOC, POC and SOC pools and we do not want the figures to be too crowded.

*Fig. 9. Do we know how sensitive are these proxies are? What are their expected compositions for marine and terrestrial OC, and how much do they vary. In other words, when we have +ve, are we talking about change of a few % of the total OC, or a lot more? And/or can the source of the OC impact the ratios. This is discussed in the main text, but the abstract and conclusions make quite assertive points based on these data.*
*See comment 1b above.*

We have now clarified throughout the manuscript that these proxies should be interpreted carefully as these products account for less than 0.1% of the bulk OC. We have also removed Figure 9A, which was representing Pn/P ratios, to prevent misunderstandings.

---

## Referee Comment (RC2) · Anonymous Referee #2 · 21 Aug 2016

The presented study provides a new and exciting data set from the outer East Siberian shelf investigating the sources and cycling of different organic carbon pools. The ongoing warming in the Arctic will likely increase the amounts of organic matter exported from land (trough rivers & coastal erosion) and our knowledge about the fate of the different terrigenous organic matter pools in the ocean and across the Siberian shelves is still surprisingly small despite recent publications trying to fill this gap. The combination of bulk dissolved, suspended particulate and sedimentary organic carbon (DOC, POC, and SOC) parameters including stable and radiocarbon isotopes as well as biomarker data (lignin phenols) from surface and bottom waters offers highly needed insights into the sources, quality, and age of the different terrigenous organic matter pools transported across the East Siberian shelf and their relationship with each other. The strong contrast between relatively young terrigenous surface water DOC and old terrigenous bottom water POC and SOC and the fact that they seem to be completely decoupled from each other are the most interesting outcomes of this study. I recommend this manuscript for publication in Biogeosciences.

In the following, you can find my questions and remarks.

**Lines 42-45:** How does the sea ice cover and Pacific inflow relate to the older and more enriched $\delta^{13}C$? Can you please elaborate here shortly?

**Line 174:** What is the pore size of the Teflon filters? Is it different from the GF/F filters and if so how could that influence the lignin yields and source as well degradation ratios in comparison to the DOC samples (filtrate from GF/F filters?)?

**Lines 183-186:** Do you have any data or reference for the fact that GF/F filters are not compatible? As far as I understand the problem is that the silica of the GF/F filters could react with the NaOH and use it up, which would change the pH and the needed alkaline conditions for the CuO oxidation. Did you ever try it? If you would use a higher concentration of NaOH, it could keep the solution in alkaline conditions despite some of the silica reacting with the NaOH.

**Lines 206-209:** Is there any estimate about the approximate age of the surface sediment slices (0.5cm and 1cm) analyzed here? The sedimentation rates are generally lower on the outer shelf than on the inner and they can vary quite strongly. Therefore, it would be important to know how much time is integrated in these samples when comparing them. Also, can you please add the information on the sediment depth of each sample in Table 3 for easier readability?

**Lines 330-335:** Is there some information/words missing here? What "processes of terrigenous DOC along the offshore transport" are reflected here? Do you want to suggest that "processes such as hydrodynamic sorting, deposition, resuspension and uptake by primary production influence the DOC $\delta^{13}C$ and $\Delta^{14}C$ concentrations? Please clarify.

**Lines 338-339**: I generally agree with the statement made here that "large proportions the DOC exported to the outer shelf comes from young vascular plant material".

Although "young" is a relative term. Based on the fact that the DOC is younger than the underlying sediment (and deep ocean DOC in the Atlantic for example), you could call it young. However, it is important to mention in this context that the DOC-$\Delta^{14}$C values of Eurasian rivers, such as the Lena, are much more enriched in $\Delta^{14}$C with values >0‰ (containing bomb-$^{14}$C, see Raymond et al. 2007, *Glob Biogeochem Cycl*) and are therefore much younger than the samples presented here. That implies that there is either a considerable change in composition (and age) within the DOC pool during the cross-shelf transport likely affecting the young and labile fraction of DOC or mixing with older marine DOC.
Please include this aspect into your discussion.

**Lines 399-401:** Here it would be helpful to add information on for example the concentration of the hexadocenoic acid (C16FA:1), which is also a product of the CuO oxidation used here, in the POC and SOC samples. As Tesi et al. (2014) have shown, the C16FA:1 is a good indicator for marine organic carbon on the East Siberian shelf. Additionally, the dual-isotope ($\delta^{13}$C, $\Delta^{14}$C) three-endmember Monte Carlo simulation (see e.g. Vonk et al., 2012) could give an estimate about the contribution of marine, ice complex and surface soil organic carbon to these samples, which would improve discussion on the fraction of terrigenous and marine OC present in these samples.

**Lines 401-403:** Please clarify that the young Terr-OC is transported mainly in the *surface* water DOC, because the statement made here does not hold true for the bottom water DOC (3 out of 4 samples are older or as old as the respective bottom water POC samples).

**Lines 431-433:** Please mention in which ways to these processes alter the original composition (e.g. degradation causing a shift to towards woody lignin).

**Lines 436-438:** Please explain in more detail here. Are you talking about the surface water or bottom water POC? These two POC pools seem to be influenced by different transport processes (more buoyant surface transport versus nepheloid layer transport, resuspension) working on different time scales (shorter at the surface versus longer, likely thousands of years, in nepheloid layer).
How would you explain the distinct clustering seen in the C/V versus S/V plot? Does selective degradation, hydrodynamic sorting, leaching/adsorption influence the source signal in the different OC pools?
Bröder et al. 2016, Biogeosciences Discussion (in your reference list still Bröder et al. submitted – please change), observed an increasing S/V ratio with increasing distance from the coast and suggested that this is rather a result of hydrodynamic sorting or selective degradation than a change in source.

Furthermore, when I looked into the data of Bröder et al. 2016 I realized that the sediment samples presented here with the IDs 1, 4, 6, 14, 23, and 24 are the same as presented there. Only here the prefix SW is missing, but the Longitude and Latitude match and as both studies are based on samples from the SWERUS-Expedition in summer 2014 I assume these are the same samples. At first, I thought you forgot to put a reference in here for the $\delta^{13}$C and lignin data of these samples. And for the $\delta^{13}$C SOC data you should do that, because Bröder et al. published it first in Biogeosciences Discussion. However, the lignin data presented here by Salvadó et al. (S/V; C/V; Sd/Sl; Vd/Vl; 3,5-Bd/V) is considerably different from the data in Bröder et al. 2016 for these samples, e.g.

in the C/V vs. S/V plot the data from Bröder et al. 2016 for these particular samples would plot much more in the direction of woody angiosperm material. So, maybe I got it all wrong and you used different samples than Bröder et al. 2016 or something got mixed up. Please check your data again and clarify.

**Lines 439-455:** First of all, the Pn/P and P/V data is missing in Tables 1, 2, and 3. Further, I'm not convinced that the Pn/P ratio is a good proxy for marine versus terrestrial OC contribution here. In Tesi et al. (2014) active layer permafrost soil samples from the Indigirka, Lena, and Kolyma watersheds yielded Pn/P ratios <0.1, which could be a possible source for the low POC values here.
As mentioned above, you could use the C16FA:1 concentrations or the dual-isotope three-endmember Monte Carlo simulation the assess the relative contributions of terrigenous and marine organic carbon in these samples.

**Line 455:** Sentence missing in the end.

**Tables 1-3:** Pn/P and P/V ratios missing.

**Tables 1 and 2:** Please explain swi in DOC-swi and POC-swi.

**Fig. 2A:** The second y-axis on the right shows a different maximum value and different intervals. Is one of the axes for POC and one for DOC? Please clarify.

**Fig. 8A:** Where do the boxes for angiosperm leaves, etc. come from? Can you give a reference?

**Fig. 9C:** It is hard to see in this figure, but is there a slight trend in SOC pCd/Fd ratio with Longitude? And if so, how would you explain that?

---

## Author Comment (AC3) · 30 Sep 2016

**Author responses to referee 2 comments, and planned edits to BG ms "*Contrasting composition of terrigenous organic matter in the dissolved, particulate and sedimentary organic carbon pools on the outer East Siberian Arctic Shelf*" (doi:10.5194/bg-2016-260, 2016)**

**by Salvadó, Tesi, Sundbom, Karlsson, Kruså, Semiletov, Panova and Gustafsson**

We thank the reviewer for her/his careful reading of our manuscript. The constructive reviews and suggestions have contributed to improve the paper further during our revisions. All referee comments and our responses, as well as the planned edits, are detailed below, organized such that first the reviewer comments are given in italic, directly followed by our response and outline of planned edits in regular font.

*Reviewer Comments*

*The presented study provides a new and exciting data set from the outer East Siberian shelf investigating the sources and cycling of different organic carbon pools. The ongoing warming in the Arctic will likely increase the amounts of organic matter exported from land (trough rivers & coastal erosion) and our knowledge about the fate of the different terrigenous organic matter pools in the ocean and across the Siberian shelves is still surprisingly small despite recent publications trying to fill this gap. The combination of bulk dissolved, suspended particulate and sedimentary organic carbon (DOC, POC, and SOC) parameters including stable and radiocarbon isotopes as well as biomarker data (lignin phenols) from surface and bottom waters offers highly needed insights into the sources, quality, and age of the different terrigenous organic matter pools transported across the East Siberian shelf and their relationship with each other. The strong contrast between relatively young terrigenous surface water DOC and old terrigenous bottom water POC and SOC and the fact that they seem to be completely decoupled from each other are the most interesting outcomes of this study.*

*I recommend this manuscript for publication in Biogeosciences.*

We appreciate this positive overall assessment of the value and suitability of our paper.

*In the following, you can find my questions and remarks.*

**Lines 42-45:** *How does the sea ice cover and Pacific inflow relate to the older and more enriched $\delta^{13}C$? Can you please elaborate here shortly?*

It is known that the Pacific inflow brings older and more enriched $\delta^{13}C$ signatures (Semiletov et al., 2005; Stein and Macdonald, 2004). In this study we also suggest that this inflow has a stronger influence under the sea ice extent, which could work as a barrier preventing the input of "young" DOC and POC.

We agree that this point may benefit from some further clarification. We have therefore now included that "the Pacific inflow and the sea ice coverage, which works as a barrier preventing the input of "young" DOC and POC, seem to have a strong influence on these carbon pools, presenting older and more enriched $\delta^{13}C$ signatures under the sea ice extent." (lines 40-43).

**Line 174:** *What is the pore size of the Teflon filters? Is it different from the GF/F filters and if so how could that influence the lignin yields and source as well degradation ratios in comparison to the DOC samples (filtrate from GF/F filters?)?*

We have now included that the pore size of the Teflon filters is 1 μm. (line 178).

We had to use Teflon filters to be able to analyse lignin-derived phenols in POC samples, as GF/F filters are not compatible with the alkaline hydrolysis of the CuO oxidation protocol. The cut-offs of Teflon and GF/F filters are as close/similar as possible and the results should be comparable. We do

not think that the Teflon filters could have influenced the different lignin yields and the degradation ratios between DOC and POC.

*Lines 183-186: Do you have any data or reference for the fact that GF/F filters are not compatible? As far as I understand the problem is that the silica of the GF/F filters could react with the NaOH and use it up, which would change the pH and the needed alkaline conditions for the CuO oxidation. Did you ever try it? If you would use a higher concentration of NaOH, it could keep the solution in alkaline conditions despite some of the silica reacting with the NaOH.*

As far as we know, there are not studies that specifically focus on how to circumvent the problem by changing the molar concentration of the NaOH solution. Winterfield et al (2015) used GF/F filters for river suspended sediments but they only used what they could scrap off the GF/F. Lobbes et al (2000) used a rotatory evaporator to concentrate the POC of the Lena River. This is of course extremely time consuming and logistically impossible for our samples (i.e. much larger volumes).

Maybe changing the molarity of NaOH could work, but it was probably safer to use an inert PTFE filter that does not dissolve or interact with the analyses.

*Lines 206-209: Is there any estimate about the approximate age of the surface sediment slices (0.5cm and 1cm) analyzed here? The sedimentation rates are generally lower on the outer shelf than on the inner and they can vary quite strongly. Therefore, it would be important to know how much time is integrated in these samples when comparing them.*

The precise age of the surface sediment slices are not known. The 8-10 [210]Pb-dated sediment cores available from ESAS indicate linear sed rates on the order of 0.5-2 mm/yr (e.g. Vonk et al., 2012; Bröder et al., 2016). Naturally, there is also some mixing in the top several centimeters. A rough estimate is that the surf sediment samples represent material sedimented over the past years to several decades.

*Also, can you please add the information on the sediment depth of each sample in Table 3 for easier readability?*

We have now also marked the three cores sectioned on high resolution (0.5 cm intervals) in Table 3.

*Lines 330-335: Is there some information/words missing here? What "processes of terrigenous DOC along the offshore transport" are reflected here? Do you want to suggest that "processes such as hydrodynamic sorting, deposition, resuspension and uptake by primary production influence the DOC $\delta^{13}C$ and $\Delta^{14}C$ concentrations? Please clarify.*

We have now clarified that both mixing and processing such as sorting and degradation of the terrigenous DOC along the off-shelf transport explain the correlation between $\delta^{13}C$ and DOC ($r^2=0.68$). Processes such as hydrodynamic sorting, deposition, resuspension and uptake by primary production may contribute to the dispersal and further fate of the OC in the ESAS. On the other hand, the relationship between $\Delta^{14}C$ and DOC ($r^2=0.87$) represents both the source of the terrigenous DOC and mixing between components. (lines 339-347).

*Lines 338-339: I generally agree with the statement made here that "large proportions the DOC exported to the outer shelf comes from young vascular plant material".*

*Although "young" is a relative term. Based on the fact that the DOC is younger than the underlying sediment (and deep ocean DOC in the Atlantic for example), you could call it young. However, it is important to mention in this context that the DOC-$\Delta^{14}C$ values of Eurasian rivers, such as the Lena, are much more enriched in $\Delta^{14}C$ with values >0‰ (containing bomb-14C, see Raymond et al. 2007, Glob Biogeochem Cycl) and are therefore much younger than the samples presented here. That implies that there is either a considerable change in composition (and age) within the DOC pool*

*during the crossshelf transport likely affecting the young and labile fraction of DOC or mixing with older marine DOC.*
*Please include this aspect into your discussion.*

We agree with this assessment and have now included that "The radiocarbon signatures in the DOC pool of the outer-shelf of the ESAS are older than those observed in the Lena River (>39‰) (Raymond et al., 2007), but younger or similar (in the outer and eastern stations) than those reported in surface waters of the Canada Basin (<-216‰) (Arctic Ocean) (Griffith et al., 2012), likely reflecting the inputs of Pacific waters. The considerable change in age within the DOC pool during the cross-shelf transport is likely due to mixing with older marine DOC." (lines 320-325).

*Lines 399-401: Here it would be helpful to add information on for example the concentration of the hexadocenoic acid (C16FA:1), which is also a product of the CuO oxidation used here, in the POC and SOC samples. As Tesi et al. (2014) have shown, the C16FA:1 is a good indicator for marine organic carbon on the East Siberian shelf. Additionally, the dual-isotope ($\delta^{13}C$, $\Delta^{14}C$) three-endmember Monte Carlo simulation (see e.g. Vonk et al., 2012) could give an estimate about the contribution of marine, ice complex and surface soil organic carbon to these samples, which would improve discussion on the fraction of terrigenous and marine OC present in these samples.*

We agree that the C16FA:1 proxy and the Monte Carlo simulations could be also useful to unravel the sources of the OC present in these samples. Unfortunately, we did not quantify that compound and we decided not to do statistical modelling in this study. We think that the extensive data presented here from the DOC and POC and SOC pools in the outer ESAS is valuable enough to elucidate interesting new issues and conclusions. Nevertheless, we agree that further studies should analyse more proxies and develop Monte Carlo simulations to strengthen some of the hypothesis and suggestions we are inferring in this study, particularly regarding sources and degradation of the DOC, POC and SOC pools in the outer ESAS.

*Lines 401-403: Please clarify that the young Terr-OC is transported mainly in the surface water DOC, because the statement made here does not hold true for the bottom water DOC (3 out of 4 samples are older or as old as the respective bottom water POC samples).*

We agree and have now clarified this issue with "whereas "young" Terr-OC is transported mainly within the surface dissolved fraction, near-bottom POC and SOC carries off-shelf preferentially old OC from remobilized permafrost." (lines 417-421)

*Lines 431-433: Please mention in which ways to these processes alter the original composition (e.g. degradation causing a shift to towards woody lignin).*

We agree and have now included that "degradation lowers both S/V and C/V ratio but largely C/V. If degradation was indeed the main process we would expect a correlation between S/V and C/V. Therefore, while some degradation cannot be excluded, the observed differences in lignin phenols between carbon pools likely reflect a different source." (lines 449-452).

*Lines 436-438: Please explain in more detail here. Are you talking about the surface water or bottom water POC? These two POC pools seem to be influenced by different transport processes (more buoyant surface transport versus nepheloid layer transport, resuspension) working on different time scales (shorter at the surface versus longer, likely thousands of years, in nepheloid layer).*

We agree with the review point; however, here we only want to differentiate the DOC, POC and SOC pools, without considering locations and water masses, as they depict clear different clustering between OC pools.

*How would you explain the distinct clustering seen in the C/V versus S/V plot? Does selective degradation, hydrodynamic sorting, leaching/adsorption influence the source signal in the different*

*OC pools? Bröder et al. 2016, Biogeosciences Discussion (in your reference list still Bröder et al. submitted – please change), observed an increasing S/V ratio with increasing distance from the coast and suggested that this is rather a result of hydrodynamic sorting or selective degradation than a change in source.*

The distinct clustering between DOC, POC and SOC pools suggests that small amounts of non-woody angiosperm tissues are mixing with large amounts of gymnosperm woods in these samples. It also suggests that angiosperms are mainly transported by SOC and gymnosperms by POC.

Bröder et al., 2016 presented a complete shelf transect in the Laptev Sea in surface sediment samples that could suggest that increasing S/V ratios were due to hydrodynamic sorting or selective degradation; however, she could not compare it with the DOC and POC pools. In this study, we are presenting samples along the outer shelf of the ESAS and we are focusing on comparing the DOC, POC and SOC pools. The classical source plot of S/V versus C/V depicts distinct clustering between DOC, POC and SOC, suggesting that SOC mainly carries angiosperms while POC is more resembling gymnosperms.

*Furthermore, when I looked into the data of Bröder et al. 2016 I realized that the sediment samples presented here with the IDs 1, 4, 6, 14, 23, and 24 are the same as presented there. Only here the prefix SW is missing, but the Longitude and Latitude match and as both studies are based on samples from the SWERUS-Expedition in summer 2014 I assume these are the same samples. At first, I thought you forgot to put a reference in here for the $\delta^{13}C$ and lignin data of these samples. And for the $d^{13}C$ SOC data you should do that, because Bröder et al. published it first in Biogeosciences Discussion. However, the lignin data presented here by Salvadó et al. (S/V; C/V; Sd/Sl; Vd/Vl; 3,5-Bd/V) is considerably different from the data in Bröder et al. 2016 for these samples, e.g. in the C/V vs. S/V plot the data from Bröder et al. 2016 for these particular samples would plot much more in the direction of woody angiosperm material. So, maybe I got it all wrong and you used different samples than Bröder et al. 2016 or something got mixed up. Please check your data again and clarify.*

We appreciate this assessment and understand the review point. It is right that these are the same samples; however, Lisa Bröder decided to inject them later altogether with the rest of her samples to minimize differences between all her samples. It is know that different injections and calibrations can affect the results a little, therefore, as the objective was to have a good comparison with DOC, POC and SOC pools, all samples presented in this manuscript were injected together. These small differences may be due to the different injections and the different calibrations used. For this reason, we did not put a reference for this data. We have now included in the manuscript that "Some sediment samples (SWE-1, SWE-4, SWE-6, SWE-14, SWE-23, SWE-24) were also analysed by Bröder et al. (2016). The small differences in lignin phenols results may stem from the different injections and calibrations used" (lines 252-255).

**Lines 439-455:** *First of all, the Pn/P and P/V data is missing in Tables 1, 2, and 3. Further, I'm not convinced that the Pn/P ratio is a good proxy for marine versus terrestrial OC contribution here. In Tesi et al. (2014) active layer permafrost soil samples from the Indigirka, Lena, and Kolyma watersheds yielded Pn/P ratios <0.1, which could be a possible source for the low POC values here.*
*As mentioned above, you could use the C16FA:1 concentrations or the dual-isotope three-endmember Monte Carlo simulation the assess the relative contributions of terrigenous and marine organic carbon in these samples.*

Sorry, we have now included Pn/P and P/V data in Tables 1, 2 and 3. We have also clarified in the manuscript that this proxy should be interpreted carefully and have now removed Figure 9A to prevent misunderstandings (line 470). As mentioned above, we did not quantify the C16FA:1 proxy and we decided not to do statistical modelling in this study.

**Line 455:** *Sentence missing in the end.*

We have now removed this text "Overall, these proxies".

***Tables 1-3:*** *Pn/P and P/V ratios missing.*

We have now added Pn/P and P/V ratios in Tables 1, 2 and 3.

***Tables 1 and 2:*** *Please explain swi in DOC-swi and POC-swi.*

We have now described -swi, seawater intake samples (surface water samples at 8 m depth); and -sub, samples obtained by submersible pump (near-bottom water samples, 5 m above bottom) in Tables 1 and 2.

***Fig. 2A:*** *The second y-axis on the right shows a different maximum value and different intervals. Is one of the axes for POC and one for DOC? Please clarify.*

We have now clarified in the figure caption that the left y-axis is for DOC and the right y-axis for POC.

***Fig. 8A:*** *Where do the boxes for angiosperm leaves, etc. come from? Can you give a reference?*

We have now added in the figure caption that "Typical ranges for woody and non-woody tissues of both angiosperm and gymnosperm vegetation are indicated as boxes in the graph (Goñi et al., 2000)."

***Fig. 9C:*** *It is hard to see in this figure, but is there a slight trend in SOC pCd/Fd ratio with Longitude? And if so, how would you explain that?*

We agree that there is also a slight trend in SOC pCd/Fd, the same observed in 3,5-Bd/V ratios. Previous studies in sediments and the colloidal fraction from the ESAS also reported the same trend (Karlsson et al., 2016; Tesi et al., 2014) reflecting the Pacific inflow from the east of more marine and degraded OC. We have now included it in the manuscript "In this data set pCd/Fd ratios follow the same pattern as the ones observed in 3,5-Bd/V ratios with higher values in POC and SOC and a slightly increasing tendency in the eastern SOC samples" (lines 526-529).

---

## Author Comment (AC4) · 30 Sep 2016

**Author responses to reviews and edits to BG ms "*Contrasting composition of terrigenous organic matter in the dissolved, particulate and sedimentary organic carbon pools on the outer East Siberian Arctic Shelf*" (doi:10.5194/bg-2016-260, 2016)**

**by Salvadó, Tesi, Sundbom, Karlsson, Kruså, Semiletov, Panova and Gustafsson**

We thank the reviewers and the Editor for their careful reading of our manuscript. We are encouraged by the overall supportive assessments. The constructive reviews and suggestions have contributed to improve the paper further during our revisions, where we have also had a special eye to improve clarity. All referee comments and our responses, as well as the resulting edits, are detailed below, organized such that first the reviewer comments are given in italic, directly followed by our response and outline of resulting edit in regular font. References in our response to line numbers refer to the new/revised ms version.

*Reviewer 1*

*Comments*

*The manuscript of Salvado et al., provides new data to examine organic carbon cycling in the East Siberian Arctic Shelf. This is an important issue. The Arctic Ocean is undergoing significant warming, and this is projected to increase over the coming century. The impact of this warming on carbon pools remains an important question – i.e. are the carbon pools stable, or could they degrade and release gaseous carbon to contribute to atmospheric $CO_2$ and $CH_4$ budgets. One of the key unknowns remains the source, age and nature of organic carbon in the water column and at the seafloor (and how these are linked). Despite a large amount of recent work in the study area (involving some of the authors), we are still sample limited, and this paper delivers an impressive new dataset.*

*A strength of the study is that they examine dissolved and particulate phases in the surface water, and deep water, alongside the sedimentary organic matter. The paper presents bulk elemental and isotopic measurements (including radiocarbon) alongside a range of biomarker yields to help reveal patterns in the source and processing of OC. These aspects allow the authors to examine the contrasting source and age of the OC, and discussion what processes are responsible for these contrasts. Age contrasts between DOC and POC have been examined before, but not too my knowledge through the water column, and in relation to the lignin biomarkers. The decoupling of deep POC and DOC is very stark. The findings are new and should certainly interest the readership at Biogeosciences.*

We appreciate this positive overall assessment of the value and suitability of our paper.

*However, I have one main comment, which is comprised of a few parallel issues/observations which I feel the authors should work to clarify in revision:*

*1. How sensitive are the biomarker proxies (used to examine marine vs terrestrial, and evidence for degradation): a) The Pn/P (hydroxyacetophenone/p-hydroxybenzoic acids) ratio is discussed in the context of a marine versus terrestrial biomarker (Fig. 9a) and used to conclude (stated in the abstract and throughout) that POC is 'mainly composed' of marine OC, suggesting more than 50% of the TOC is marine in origin. However, I think the paper needs to be a little more cautious on this point. They discuss some of the caveats of this biomarker ratio in the text, but also remember this is an extracted component of the TOC.*

We understand the review point and have now clarified these points throughout the manuscript (lines 461-464, 468-475). We do not mean to base conclusions only on this Pn/P ratio. We have thus removed Figure 9A to prevent misunderstandings. We have now edited to state that POC in surface waters may have a preferential marine origin, mainly inferred from its very low lignin concentrations, compared to DOC and SOC pools (Figures 2B and 4). Further, we have also corrected that the Pn/P ratios observed in DOC (0.08-0.37), SOC (0.06-0.17) and POC (0.02-0.14) suggest that the OC in the

dissolved pool is more terrestrial than in the particulate and sediment pools (Tables 1, 2 and 3; Figure 9A). Nevertheless, these proxies should be interpreted carefully as P products account for less than 0.1% of the bulk OC so care should be placed when extrapolating the biomarker data (lines 468-475).

*What are the bulk proxies (elemental, isotope) telling us? Indeed, what is the lignin yield telling us? After two reads, to me, the lignin yield data doesn't seem to line up with this conclusion. Some surface POC has more lignin yield than the DOC (and not significantly less) and the deep POC samples also have high Lignin yields. Have I missed something here? I don't follow how the lignin yield (a 'major' biomarker if you like) can show these patterns, but the bulk Pn/P ratio tell us something else about organic matter source? The POC is 13C-depleted (note – seems to correlate with DOC d13C looking at the maps) which could be marine organic matter soured from DIC with a remineralisation signature (as the authors explain somewhere). Or it could be the 'top soil' end member identified by Vonk et al., 2012, Nature, at -26 to -30 per mil (see the supplement). Added to this, the 14C-depleted nature of the bulk, deep water POC is easier to explain if a 'permafrost' terrestrial OC signature, so again, how can this be material dominated by marine OC?*

We agree and have now clarified these matters (lines 364-368, 470-475). The bulk proxies and the lignin yield are telling us that there is a significant decoupling of terrestrial OC sources and pathways. While higher lignin values in "young" DOC are related to the fluvial discharge of fresh OC from the Lena River, enhanced lignin yields in "old" POC, from near-bottom waters, most likely reflect the off-shelf transport of permafrost-derived OC in the nepheloid layer (Figures 2B, 3 and 4).

We have now included in the manuscript that some surface POC samples, the ones in the Kara Sea, had more lignin than in the DOC, but those concentrations were relatively low compared to the higher lignin yields observed in the dissolved and sedimentary OC pools from the ESAS (lines 364-368). We also explain in many parts of the manuscript that the enhanced lignin values in near-bottom waters may be due to the off-shelf transport of permafrost-derived OC in the nepheloid layer, through repeated cycles of deposition and resuspension across the shelf (lines 47-49, 329-331, 339-341, 381-383, 412-414, 419-421, 551-553, 566-568).

We agree that the Pn/P ratio is not following the same trend as the lignin yield in the particulate and sedimentary OC pools. We have thus clarified that this proxy should be interpreted carefully and removed Figure 9A to prevent misunderstandings (lines 472-473). On the other hand, we are confident stating that POC in surface waters of the outer-shelf of the ESAS is composed preferentially of marine OC mainly due to its very low lignin concentrations, compared to DOC and SOC pools (Figures 2 and 4), and its no-correlations with $\delta^{13}$C and lignin values.

*b) I have a related comment on the degradation proxies. I think these are important to pursue, so they should certainly be in here and discussed. But, there are some key questions. Do we know the starting compositions of organic matter? The authors acknowledge we have no POC samples from terrestrial materials to compare too. Second, if the biomarkers suggest 'significant' degradation (or as the authors put it, more than we have seen before), should we not also see this in the bulk %OC too? Is there a link in this dataset, or is the %OC too variable due to other factors (sorting, heterogeneity). Indeed, for the SOC samples, a quick look in Table 3 reveals those with high 3,5Bd/V (1.87, 1.89, 2.38) appear to have relatively high %OC (1.7%, 1.3%, 1,7% respectively), while those with lower %OC (0.4%, 0.5%) have lower 3,5Bd/V (0.72, 0.59). This is the opposite of what you might expect for degradation, i.e. higher 3,5Bd/V should indicate more degradation and lower %OC. Does this suggest a strong role of a source change, rather than a degradation signal, to explain the 3,5Bd/V ratios?*

We understand the review point; however, we cannot assume that higher 3,5Bd/V necessarily correlates with lower %OC. For example, OC can be affected by several other factors which include grain-size and dilution with marine OC. Therefore, the OC content itself does not necessarily reflect the extent of degradation as the 3,5Bd/V ratio does.

Nevertheless, we agree and have now included in the text that 3,5Bd/V could support the role of a source change as observed with S/V and C/V proxies (lines 510-513). In this data set pCd/Fd ratios, another ratio that has been used as a diagenetic indicator (Amon et al., 2012; Houel et al., 2006), followed the same pattern as the ones observed in 3,5-Bd/V ratios with higher values in POC and SOC (Tables 1, 2 and 3; Figure 9C). This would strengthen the hypothesis that POC and SOC are more degraded than DOC (lines 526-529).

*More discussion on these issues, and awareness of the caveats would benefit the manuscript. I wonder if a combined results/discussion is helping this. It could be wise to separate these sections to clarify some of these points.*

We have now strengthened the discussion of these aspects in the manuscript.

*Other comments (by line numbers)*

*37: based on my reading of the manuscript and the data, a key contrast is that the lignin is highest in the oldest POC samples, whereas the lignin is highest in the youngest DOC samples, suggesting a significant decoupling of terrestrial OC sources and pathways. I was less convinced by the evidence for 'preferential' (implies >50% ?) marine origin of the POC (especially at the seafloor).*

We agree with this assessment and have now deleted this sentence here in the abstract. Moreover, we have now clarified throughout the manuscript that POC in surface waters may have a preferential marine origin mainly due to its very low lignin values. We also state now in many parts of the manuscript that the higher lignin values in younger DOC samples are likely related to the fluvial discharge of fresh OC from the Lena River, while the higher lignin yields in older POC samples, from near-bottom waters, is bound to reflect the off-shelf transport of permafrost-derived OC in the nepheloid layer.

*44: define ESS*

We have now deleted this acronym as we have restructured the abstract.

*36, 47 & 58: these are important observations, but they are repeated here. Consider restructuring the abstract to make the results and implications/discussion more distinct.*

We agree and have now restructured the abstract. We have now deleted the sentences "Depleted $\delta^{13}$C, modern $\Delta^{14}$C and lignin phenols concentrations were all well correlated with DOC levels indicating a relatively young terrestrial contribution. In contrast, POC may have a preferential marine origin, as its concentrations were not correlated with isotope and terrestrial biomarker proxies" and "Overall, DOC is strongly affected by the Lena River plume transporting young Terr-OC from topsoil and/or recently produced vascular pant material, while near-bottom POC and SOC preferentially carries off-shelf old OC released from thawing permafrost". Further, we have now included the sentence "Overall, the key contrast between enhanced lignin yields both in the youngest DOC and the oldest POC samples suggest a significant decoupling of terrestrial OC sources and pathways" (lines 54-56).

*45: the sea ice observation (Fig. 4) seems important, but it is not well explained. Perhaps reorganise the abstract and add a sentence to more clearly make this point.*

We have now restructured the abstract and rewritten this sentence "Further, the Pacific inflow and the sea ice coverage, which works as a barrier preventing the input of "young" DOC and POC, seem to have a strong influence in these carbon pools, presenting older and more enriched $\delta^{13}$C signatures under the sea ice extent" (lines 40-43).

*57: how much evidence for this is presented? Perhaps add a caveat about lignin yields not being that different, and %OC being not that different (more bulk indicators of net degradation?)?*

We agree that there in not a strong evidence for this; therefore, we have now deleted this sentence: "This suggests that the remobilized old OC from thawing permafrost, which is mainly transported within these pools, could experience less burial and more mineralization than believed earlier." Moreover, we have included the sentence "suggesting more degradation within these pools". (line 54).

*60: ended on a rather generic note. The final sentences could do a better job of summing up the main findings.*

We have now rewritten this sentence "Overall, the key contrast between enhanced lignin yields both in the youngest DOC and the oldest POC samples reflects a significant decoupling of terrestrial OC sources and pathways" (lines 54-56).

*87: there are two issues here. 1st the potential fate of OC at the seafloor due to warming (we know nothing about this?!) and 2nd the changing terrestrial inputs. The text could be modified to make these points.*

We have now included these points: "The sources and the inputs of Terr-OC are likely to vary in the northern shelf margin due to the changing climate" (lines 86-87).

*100: define acronyms at first use here.*

We have now defined DOC, POC and SOC here (lines 100-101).

*109: Line 102 mentions the Arctic Ocean here, but I think these comments are linked to the Eurasian Shelf. Certainly, the POC does not appear to degrade faster than POC in the beaufort Sea/Mackenzie Delta studies, probably because of the much higher sediment input at this point. Anyhow, worth clarifying.*

This appears to be a misunderstanding. Here we are referring to studies in different places of the Arctic Ocean that indicated a conservative behavior of DOC.

*116: another issue worth flagging is that the POC in the Eurasian rivers is not well characterised. Previous studies (Vonk et al and others) rely on estuarine samples.*

We have now clarified that POC in the Eurasian rivers is not well characterised (lines 121-122).

*127: could be useful to provide estimates of these OC fluxes.*

We have now included an estimate of Terr-OC input in the ESAS "(22±8 Tg OC/yr; Vonk et al., 2012)" (lines 128-129).

*137: I agree, this is cool, but could you better explain why it is important to do this.*

We have now explained why it is important to do this study "…seeking to test the hypothesis that carbon pools may carry different types of OC with different propensity toward off-shelf transport and degradation" (lines 140-142).

*153: how much? (% or Tg C yr-1)?*

We have now included the percentage of ICD-PF that enters the Laptev Sea (53±5%; Vonk et al., 2012) (line 158).

*201: to what depth was the sediment sampled? I wasn't too clear on whether these were surface samples, or cores*

We have now clarified that "Sediment cores were collected from 40 to 3120 m water depth" and that "This study focused on surface sediments (0-1 cm)." (lines 206 and 215).

*221: HCl fumigation (vapour) or leach (liquid) method?*

We have now clarified that "surface sediment samples were subsampled and subject to aqueous acidification with HCl (1.5M)" (line 225).

*304: this DOC D14C is much older than reported in the study of Eurasian Arctic Rivers by Raymond et al., 2007, GBC, who have values D14C > +39 per mil in the Lena! Worth discussion.*

We agree and have now included "The radiocarbon signatures in the DOC pool of the outer-shelf of the ESAS are much older than those observed in the Lena River (>39‰) (Raymond et al., 2007), but younger or similar (in the outer and eastern stations) than those reported in surface waters of the Canada Basin (<-216‰) (Arctic Ocean) (Griffith et al, 2012), likely reflecting the inputs of Pacific waters." (lines 320-325).

*322-327: This seems like an important observation and worth expanding on (certainly worth its own paragraph). To me, this is the key evidence for decoupling of the POC and DOC pools, and as the authors point out, the evidence for pre-aged OC in the system.*

We have now expanded these statements "It is important to point out that near-bottom waters presented more depleted and similar $\Delta^{14}$C signatures in both $DOC_{SPE}$ and POC (-258±94‰ and -250±83‰, respectively) than in surface waters (-213±93‰ and -57±86‰, respectively) (Figure 5; Tables 1 and 2), suggesting the same older and terrigenous source of OC in both pools. Those contrasting age offsets between surface and near-bottom waters, particularly for the POC fraction, may reflect the off-shelf transport of OC translocated over long distances from thawing permafrost" (lines 339-341).

*328-339: this discussion needs to be more clear on whether you are invoking mixing to explain the patterns, or processes, or both.*

We have now clarified this discussion stating that both mixing and processes of the terrigenous DOC along the off-shelf transport explain the correlation between $\delta^{13}$C and DOC ($r^2$=0.68). Further, the relationship between $\Delta^{14}$C and DOC ($r^2$=0.87) represents both the source of the terrigenous DOC and mixing between components (lines 344-352).

*355: sentence could be clearer.*

We have now clarified this sentence "Lignin concentrations exhibited contrasting offsets between surface and near-bottom waters, particularly in the POC pool" (lines 371-373).

*364-366: interesting. How does this work hydrodynamically? Lignin rich materials may contain more coarser woody particles, which should be buoyant/neutrally buoyant. This suggests not, and indicates waterlogged terr-OC which can sink. This is seen in the Mackenzie River (but we don't have samples from the Eurasian rivers to examine).*

We appreciate this assessment; however, vascular plant debris does not float when vascular plant gets old. Actually, it has a high settling velocity because of its dimension (i.e. coarse) (see Tesi et al., 2016). Wood floats because of its air inside that fills all the conducts (e.g., xylem and floem vessels). An old debris floats until its vessels are replaced with water, then it sinks and behaves hydraulically as sandy material. The lignin we see in the outer-shelf is adsorbed on the fine fraction traveling across the margin (see Tesi et al, 2016).

*374: 'addition' rather than 'dilution'?*

We think it is better to state "concentrations in DOC, POC and SOC pools may be interpreted to result from dilution with marine organic during transport" (lines 390-393).

*394: split the paragraph here. These are important observations which need to be better drawn out.*

We have now split the paragraph into two (line 411).

*437-438: this sentence doesn't fit well with the discussion on 480, where V is noted to be sensitive to degradation. So, could the shift in S/V space for the POC be due to this, and not changing source? Why is the sediment not V depleted?*

We understand the review point; however, the ratio cinnamyl over vanillyl (C/V) is just used to distinguish woody lignin from other sources (Goni and Hedges, 1992; Hedges and Mann, 1979). We have now clarified that "degradation lowers both S/V and C/V ratio but largely C/V. If degradation was indeed the main process we would expect a sort of linear correlation between S/V and C/V. Therefore, while some degradation cannot be denied, differences in lignin phenols between carbon pools likely reflect a different source" (lines 449-452).

*454: I don't understand how this can be the case ('mainly composed' implies that >50% OC is marine) when the lignin yields (Fig 2) are higher in some POC samples than for DOC. Doesn't that suggest the same lignin loading, and similar contributions from terrestrial OC in POC and DOC?*

We agree and have now rewritten these statements: "These ratios are in agreement with the relationships of DOC and $\Delta^{14}$C and $\delta^{13}$C presented above, which indicate that the DOC exported off-shelf is mainly "young" and terrestrial. Nevertheless, it is important to note that these proxies should be interpreted carefully as P products account for less than 0.1-0.2% of the bulk OC, and in some samples the particulate and sedimentary pools have more lignin yield than the dissolved OC" (lines 470-475).

*455 & 526: role of sea ice and pacific inflow – when I read these bits of text, I thought it was not well supported by the data. After a second reading and closer look at Figure 4, I see your point. This needed to be better explained (why would sea ice do this?) and perhaps drawn out to give this its own paragraph in the discussion.*

We have now better explained the role of the sea ice and the pacific inflow: "The older and enriched $\delta^{13}$C signatures in the outer-shelf of the ESS may reflect the influence of sea ice coverage and the Pacific inflow from the East. We suggest that the sea ice would work as a barrier preventing the direct terrigenous input from inland and reinforcing the influence of Pacific waters" (lines 313-317). The influence of the Pacific inflow from the East has also been observed in several other studies (i.e. Semiletov et al., 2005; Stein and Macdonald, 2004; Karlsson et al., 2015).

*455: missing text*

We have now removed this text "Overall, these proxies" (line 475).

*472: are these Arctic studies or global – clarify.*

We have now clarified that these are global studies (lines 490-491).

*477- how sensitive are these ratios? are they a linear function of degradation? The second point is important when interpreting what a change from 1-3 actually means.*

We use these ratios just to estimate which carbon pool is more degraded. They are not a linear function of degradation. Further, these ratios should be interpreted carefully as these products account for less than 0.1% of the bulk OC.

*519-: Overall I found the conclusions didn't map as well onto the discussion and data as they could have done.*

We have now rewritten some parts of the conclusions section:

"Near-bottom waters present more depleted $\Delta^{14}C$ signatures and higher concentrations of lignin, particularly within the POC pool. This is a key evidence of decoupling of the POC and DOC pools and reflects the off-shelf transport of permafrost-derived OC in the nepheloid layer, through repeated cycles of deposition and resuspension across the shelf" (lines 549-553).

"Moreover, the opposite trends in the Pn/P and P/V ratios also indicate the terrigenous source of DOC" (lines 556-557).

"If this hypothesis is true, the remobilized OC from permafrost, which is mainly transported within these carbon pools, could experience less burial and more mineralization than the DOC pool" (lines 562-564).

"The high abundance of Terr-OC in the outer ESAS, particularly in the dissolved and sedimentary carbon pools, is a clear indicator of the magnitude of shelf to basin transport. Overall, the results are a key evidence for decoupling of the DOC, POC and SOC pools in the ESAS and elucidate the off-shelf transport of permafrost-derived OC in the particulate pool of near-bottom waters" (lines 564-568).

*527: seemed to be a somewhat misleading statement based on the available data. It would be better to explain the key findings first, and discussion what they indicate, with caveats.*

We agree and have now removed this statement. We have also rewritten the following sentences: "Near-bottom waters present more depleted $\Delta^{14}C$ signatures and higher concentrations of lignin, particularly within the POC pool. This is a key evidence for decoupling of the POC and DOC pools and reflects the off-shelf transport of permafrost-derived OC in the nepheloid layer, through repeated cycles of deposition and resuspension across the shelf" (lines 549-553).

*531: the S/V ratios show a mixture in fig. 8a? not a dominant source from one end member??*

The ratios of S/V indicate gymnosperm vegetation as the most important source of lignin (Figure 8A), and increasing S/V ratios in the easternmost samples reflect a relatively higher source contribution of tundra plants.

*541: what do you mean by 'less' burial and 'more' mineralisation. Vonk et al., (and others) already indicate this is going on. So, do you mean the numbers could be wrong by a lot (i.e. 50%) or a little (1-5%), or we don't know.*

We have now clarified that regarding 3,5-Bd/V and pCd/Fd ratios, which are higher in the SOC and POC pools, the remobilized OC from permafrost (mainly transported within these carbon pools) could experience less burial and more mineralization than the DOC pool (lines 562-564).

*Fig 2b – my comment about lignin yields is contained here (1a above). Subwater and surface POC can have higher lignin yields than DOC, but the main text then concludes a mainly marine composition for POC – I don't follow that logic.*

We have now clarified throughout the manuscript that some surface POC samples, the ones in the Kara Sea, had a larger lignin fraction than that in the DOC, but those Kara Sea lignin concentrations were relatively low compared to the higher lignin yields observed in the dissolved and sedimentary OC pools from the ESAS.

*Fig 5- nice data. Could you plot (a) and (b), with DOC one lot, and POC on the other, just to help see the surface-depth contrast in the POC.*
We have now plotted $\Delta^{14}$C of POC and SOC in Figure 5A and DOC and SOC in Figure 5B.

*Fig 6 – why exponential functions?*
We observed that our points followed exponential functions.

*Fig. 8. Plot the surface/subsurface distinction here.*

We prefer not to modify these figures. We only want to differentiate the DOC, POC and SOC pools and we do not want the figures to be too crowded.

*Fig. 9. Do we know how sensitive are these proxies are? What are their expected compositions for marine and terrestrial OC, and how much do they vary. In other words, when we have +ve, are we talking about change of a few % of the total OC, or a lot more? And/or can the source of the OC impact the ratios. This is discussed in the main text, but the abstract and conclusions make quite assertive points based on these data.*
*See comment 1b above.*

We have now clarified throughout the manuscript that these proxies should be interpreted carefully as these products account for less than 0.1% of the bulk OC. We have also removed Figure 9A, which was representing Pn/P ratios, to prevent misunderstandings.

*Reviewer 2*

*Reviewer Comments*

*The presented study provides a new and exciting data set from the outer East Siberian shelf investigating the sources and cycling of different organic carbon pools. The ongoing warming in the Arctic will likely increase the amounts of organic matter exported from land (trough rivers & coastal erosion) and our knowledge about the fate of the different terrigenous organic matter pools in the ocean and across the Siberian shelves is still surprisingly small despite recent publications trying to fill this gap. The combination of bulk dissolved, suspended particulate and sedimentary organic carbon (DOC, POC, and SOC) parameters including stable and radiocarbon isotopes as well as biomarker data (lignin phenols) from surface and bottom waters offers highly needed insights into the sources, quality, and age of the different terrigenous organic matter pools transported across the East Siberian shelf and their relationship with each other. The strong contrast between relatively young terrigenous surface water DOC and old terrigenous bottom water POC and SOC and the fact that they seem to be completely decoupled from each other are the most interesting outcomes of this study.*

*I recommend this manuscript for publication in Biogeosciences.*

We appreciate this positive overall assessment of the value and suitability of our paper.

*In the following, you can find my questions and remarks.*

***Lines 42-45:*** *How does the sea ice cover and Pacific inflow relate to the older and more enriched $\delta^{13}$C? Can you please elaborate here shortly?*

It is known that the Pacific inflow brings older and more enriched $\delta^{13}$C signatures (Semiletov et al., 2005; Stein and Macdonald, 2004). In this study we also suggest that this inflow has a stronger influence under the sea ice extent, which could work as a barrier preventing the input of "young" DOC and POC.

We agree that this point may benefit from some further clarification. We have therefore now included that "the Pacific inflow and the sea ice coverage, which works as a barrier preventing the input of

"young" DOC and POC, seem to have a strong influence on these carbon pools, presenting older and more enriched $\delta^{13}$C signatures under the sea ice extent." (lines 40-43).

**Line 174:** *What is the pore size of the Teflon filters? Is it different from the GF/F filters and if so how could that influence the lignin yields and source as well degradation ratios in comparison to the DOC samples (filtrate from GF/F filters?)?*

We have now included that the pore size of the Teflon filters is 1µm. (line 178).

We had to use Teflon filters to be able to analyse lignin-derived phenols in POC samples, as GF/F filters are not compatible with the alkaline hydrolysis of the CuO oxidation protocol. The cut-offs of Teflon and GF/F filters are as close/similar as possible and the results should be comparable. We do not think that the Teflon filters could have influenced the different lignin yields and the degradation ratios between DOC and POC.

**Lines 183-186:** *Do you have any data or reference for the fact that GF/F filters are not compatible? As far as I understand the problem is that the silica of the GF/F filters could react with the NaOH and use it up, which would change the pH and the needed alkaline conditions for the CuO oxidation. Did you ever try it? If you would use a higher concentration of NaOH, it could keep the solution in alkaline conditions despite some of the silica reacting with the NaOH.*

As far as we know, there are not studies that specifically focus on how to circumvent the problem by changing the molar concentration of the NaOH solution. Winterfield et al (2015) used GF/F filters for river suspended sediments but they only used what they could scrap off the GF/F. Lobbes et al (2000) used a rotatory evaporator to concentrate the POC of the Lena River. This is of course extremely time consuming and logistically impossible for our samples (i.e. much larger volumes).

Maybe changing the molarity of NaOH could work, but it was probably safer to use an inert PTFE filter that does not dissolve or interact with the analyses.

**Lines 206-209:** *Is there any estimate about the approximate age of the surface sediment slices (0.5cm and 1cm) analyzed here? The sedimentation rates are generally lower on the outer shelf than on the inner and they can vary quite strongly. Therefore, it would be important to know how much time is integrated in these samples when comparing them.*

The precise age of the surface sediment slices are not known. The 8-10 $^{210}$Pb-dated sediment cores available from ESAS indicate linear sed rates on the order of 0.5-2 mm/yr (e.g. Vonk et al., 2012; Bröder et al., 2016). Naturally, there is also some mixing in the top several centimeters. A rough estimate is that the surf sediment samples represent material sedimented over the past years to several decades.

*Also, can you please add the information on the sediment depth of each sample in Table 3 for easier readability?*

We have now also marked the three cores sectioned on high resolution (0.5 cm intervals) in Table 3.

**Lines 330-335:** *Is there some information/words missing here? What "processes of terrigenous DOC along the offshore transport" are reflected here? Do you want to suggest that "processes such as hydrodynamic sorting, deposition, resuspension and uptake by primary production influence the DOC $\delta^{13}$C and $\Delta^{14}$C concentrations? Please clarify.*

We have now clarified that both mixing and processing such as sorting and degradation of the terrigenous DOC along the off-shelf transport explain the correlation between $\delta^{13}$C and DOC ($r^2$=0.68). Processes such as hydrodynamic sorting, deposition, resuspension and uptake by primary production may contribute to the dispersal and further fate of the OC in the ESAS. On the other hand,

the relationship between $\Delta^{14}C$ and DOC ($r^2$=0.87) represents both the source of the terrigenous DOC and mixing between components. (lines 339-347).

***Lines 338-339:*** *I generally agree with the statement made here that "large proportions the DOC exported to the outer shelf comes from young vascular plant material".*

*Although "young" is a relative term. Based on the fact that the DOC is younger than the underlying sediment (and deep ocean DOC in the Atlantic for example), you could call it young. However, it is important to mention in this context that the DOC-$\Delta^{14}C$ values of Eurasian rivers, such as the Lena, are much more enriched in $\Delta^{14}C$ with values >0‰ (containing bomb-14C, see Raymond et al. 2007, Glob Biogeochem Cycl) and are therefore much younger than the samples presented here. That implies that there is either a considerable change in composition (and age) within the DOC pool during the crossshelf transport likely affecting the young and labile fraction of DOC or mixing with older marine DOC.*
*Please include this aspect into your discussion.*

We agree with this assessment and have now included that "The radiocarbon signatures in the DOC pool of the outer-shelf of the ESAS are older than those observed in the Lena River (>39‰) (Raymond et al., 2007), but younger or similar (in the outer and eastern stations) than those reported in surface waters of the Canada Basin (<-216‰) (Arctic Ocean) (Griffith et al., 2012), likely reflecting the inputs of Pacific waters. The considerable change in age within the DOC pool during the cross-shelf transport is likely due to mixing with older marine DOC." (lines 320-325).

***Lines 399-401:*** *Here it would be helpful to add information on for example the concentration of the hexadocenoic acid (C16FA:1), which is also a product of the CuO oxidation used here, in the POC and SOC samples. As Tesi et al. (2014) have shown, the C16FA:1 is a good indicator for marine organic carbon on the East Siberian shelf. Additionally, the dual-isotope ($\delta^{13}C$, $\Delta^{14}C$) three-endmember Monte Carlo simulation (see e.g. Vonk et al., 2012) could give an estimate about the contribution of marine, ice complex and surface soil organic carbon to these samples, which would improve discussion on the fraction of terrigenous and marine OC present in these samples.*

We agree that the C16FA:1 proxy and the Monte Carlo simulations could be also useful to unravel the sources of the OC present in these samples. Unfortunately, we did not quantify that compound and we decided not to do statistical modelling in this study. We think that the extensive data presented here from the DOC and POC and SOC pools in the outer ESAS is valuable enough to elucidate interesting new issues and conclusions. Nevertheless, we agree that further studies should analyse more proxies and develop Monte Carlo simulations to strengthen some of the hypothesis and suggestions we are inferring in this study, particularly regarding sources and degradation of the DOC, POC and SOC pools in the outer ESAS.

***Lines 401-403:*** *Please clarify that the young Terr-OC is transported mainly in the surface water DOC, because the statement made here does not hold true for the bottom water DOC (3 out of 4 samples are older or as old as the respective bottom water POC samples).*

We agree and have now clarified this issue with "whereas "young" Terr-OC is transported mainly within the surface dissolved fraction, near-bottom POC and SOC carries off-shelf preferentially old OC from remobilized permafrost." (lines 417-421)

***Lines 431-433:*** *Please mention in which ways to these processes alter the original composition (e.g. degradation causing a shift to towards woody lignin).*

We agree and have now included that "degradation lowers both S/V and C/V ratio but largely C/V. If degradation was indeed the main process we would expect a correlation between S/V and C/V. Therefore, while some degradation cannot be excluded, the observed differences in lignin phenols between carbon pools likely reflect a different source." (lines 449-452).

*Lines 436-438: Please explain in more detail here. Are you talking about the surface water or bottom water POC? These two POC pools seem to be influenced by different transport processes (more buoyant surface transport versus nepheloid layer transport, resuspension) working on different time scales (shorter at the surface versus longer, likely thousands of years, in nepheloid layer).*

We agree with the review point; however, here we only want to differentiate the DOC, POC and SOC pools, without considering locations and water masses, as they depict clear different clustering between OC pools.

*How would you explain the distinct clustering seen in the C/V versus S/V plot? Does selective degradation, hydrodynamic sorting, leaching/adsorption influence the source signal in the different OC pools? Bröder et al. 2016, Biogeosciences Discussion (in your reference list still Bröder et al. submitted – please change), observed an increasing S/V ratio with increasing distance from the coast and suggested that this is rather a result of hydrodynamic sorting or selective degradation than a change in source.*

The distinct clustering between DOC, POC and SOC pools suggests that small amounts of non-woody angiosperm tissues are mixing with large amounts of gymnosperm woods in these samples. It also suggests that angiosperms are mainly transported by SOC and gymnosperms by POC.

Bröder et al., 2016 presented a complete shelf transect in the Laptev Sea in surface sediment samples that could suggest that increasing S/V ratios were due to hydrodynamic sorting or selective degradation; however, she could not compare it with the DOC and POC pools. In this study, we are presenting samples along the outer shelf of the ESAS and we are focusing on comparing the DOC, POC and SOC pools. The classical source plot of S/V versus C/V depicts distinct clustering between DOC, POC and SOC, suggesting that SOC mainly carries angiosperms while POC is more resembling gymnosperms.

*Furthermore, when I looked into the data of Bröder et al. 2016 I realized that the sediment samples presented here with the IDs 1, 4, 6, 14, 23, and 24 are the same as presented there. Only here the prefix SW is missing, but the Longitude and Latitude match and as both studies are based on samples from the SWERUS-Expedition in summer 2014 I assume these are the same samples. At first, I thought you forgot to put a reference in here for the $\delta^{13}C$ and lignin data of these samples. And for the $d^{13}C$ SOC data you should do that, because Bröder et al. published it first in Biogeosciences Discussion. However, the lignin data presented here by Salvadó et al. (S/V; C/V; Sd/Sl; Vd/Vl; 3,5-Bd/V) is considerably different from the data in Bröder et al. 2016 for these samples, e.g. in the C/V vs. S/V plot the data from Bröder et al. 2016 for these particular samples would plot much more in the direction of woody angiosperm material. So, maybe I got it all wrong and you used different samples than Bröder et al. 2016 or something got mixed up. Please check your data again and clarify.*

We appreciate this assessment and understand the review point. It is right that these are the same samples; however, Lisa Bröder decided to inject them later altogether with the rest of her samples to minimize differences between all her samples. It is know that different injections and calibrations can affect the results a little, therefore, as the objective was to have a good comparison with DOC, POC and SOC pools, all samples presented in this manuscript were injected together. These small differences may be due to the different injections and the different calibrations used. For this reason, we did not put a reference for this data. We have now included in the manuscript that "Some sediment samples (SWE-1, SWE-4, SWE-6, SWE-14, SWE-23, SWE-24) were also analysed by Bröder et al. (2016). The small differences in lignin phenols results may stem from the different injections and calibrations used" (lines 252-255).

*Lines 439-455: First of all, the Pn/P and P/V data is missing in Tables 1, 2, and 3. Further, I'm not convinced that the Pn/P ratio is a good proxy for marine versus terrestrial OC contribution here. In*

*Tesi et al. (2014) active layer permafrost soil samples from the Indigirka, Lena, and Kolyma watersheds yielded Pn/P ratios <0.1, which could be a possible source for the low POC values here. As mentioned above, you could use the C16FA:1 concentrations or the dual-isotope three-endmember Monte Carlo simulation the assess the relative contributions of terrigenous and marine organic carbon in these samples.*

Sorry, we have now included Pn/P and P/V data in Tables 1, 2 and 3. We have also clarified in the manuscript that this proxy should be interpreted carefully and have now removed Figure 9A to prevent misunderstandings (line 470). As mentioned above, we did not quantify the C16FA:1 proxy and we decided not to do statistical modelling in this study.

*Line 455: Sentence missing in the end.*

We have now removed this text "Overall, these proxies".

*Tables 1-3: Pn/P and P/V ratios missing.*

We have now added Pn/P and P/V ratios in Tables 1, 2 and 3.

*Tables 1 and 2: Please explain swi in DOC-swi and POC-swi.*

We have now described -swi, seawater intake samples (surface water samples at 8 m depth); and -sub, samples obtained by submersible pump (near-bottom water samples, 5 m above bottom) in Tables 1 and 2.

*Fig. 2A: The second y-axis on the right shows a different maximum value and different intervals. Is one of the axes for POC and one for DOC? Please clarify.*

We have now clarified in the figure caption that the left y-axis is for DOC and the right y-axis for POC.

*Fig. 8A: Where do the boxes for angiosperm leaves, etc. come from? Can you give a reference?*

We have now added in the figure caption that "Typical ranges for woody and non-woody tissues of both angiosperm and gymnosperm vegetation are indicated as boxes in the graph (Goñi et al., 2000)."

*Fig. 9C: It is hard to see in this figure, but is there a slight trend in SOC pCd/Fd ratio with Longitude? And if so, how would you explain that?*

We agree that there is also a slight trend in SOC pCd/Fd, the same observed in 3,5-Bd/V ratios. Previous studies in sediments and the colloidal fraction from the ESAS also reported the same trend (Karlsson et al., 2016; Tesi et al., 2014) reflecting the Pacific inflow from the east of more marine and degraded OC. We have now included it in the manuscript "In this data set pCd/Fd ratios follow the same pattern as the ones observed in 3,5-Bd/V ratios with higher values in POC and SOC and a slightly increasing tendency in the eastern SOC samples" (lines 526-529).